# DEMI: DISCRIMINATIVE ESTIMATOR OF MUTUAL INFORMATION

## ABSTRACT

Estimating mutual information between continuous random variables is often intractable and extremely challenging for high-dimensional data. Recent progress has leveraged neural networks to optimize variational lower bounds on mutual information. Although showing promise for this difficult problem, the variational methods have been theoretically and empirically proven to have serious statistical limitations: 1) many methods struggle to produce accurate estimates when the underlying mutual information is either low or high; 2) the resulting estimators may suffer from high variance. Our approach is based on training a classifier that provides the probability that a data sample pair is drawn from the joint distribution rather than from the product of its marginal distributions. Moreover, we establish a direct connection between mutual information and the average log odds estimate produced by the classifier on a test set, leading to a simple and accurate estimator of mutual information. We show theoretically that our method and other variational approaches are equivalent when they achieve their optimum, while our method sidesteps the variational bound. Empirical results demonstrate high accuracy of our approach and the advantages of our estimator in the context of representation learning.

## 1 INTRODUCTION

Mutual information (MI) measures the information that two random variables share. MI quantifies the statistical dependency — linear and non-linear — between two variables. This property has made MI a crucial measure in machine learning. In particular, recent work in unsupervised representation learning has built on optimizing MI between latent representations and observations (Chen et al., 2016; Zhao et al., 2018; Oord et al., 2018; Hjelm et al., 2018; Tishby & Zaslavsky, 2015; Alemi et al., 2018; Ver Steeg & Galstyan, 2014). Maximization of MI has long been a default method for multi-modality image registration (Maes et al., 1997), especially in medical applications (Wells III et al., 1996), though in most work the dimensionality of the random variables is very low. Here, coordinate transformations on images are varied to maximize their MI.

Estimating MI from finite data samples has been challenging and is intractable for most continuous probabilistic distributions. Traditional MI estimators (Suzuki et al., 2008; Darbellay & Vajda, 1999; Kraskov et al., 2004; Gao et al., 2015) do not scale well to modern machine learning problems with high-dimensional data. This impediment has motivated the construction of variational bounds for MI (Nguyen et al., 2010; Barber & Agakov, 2003); in recent years this has led to maximization procedures that use deep learning architectures to parameterize the space of functions, exploiting the expressive power of neural networks (Song & Ermon, 2019; Belghazi et al., 2018; Oord et al., 2018; Mukherjee et al., 2020).

Unfortunately, optimizing lower bounds on MI has serious statistical limitations. Specifically, McAllester & Stratos (2020) showed that any high-confidence distribution-free lower bound cannot exceed $O(\log N)$, where N is the number of samples. This implies that if the underlying MI is high, it cannot be accurately and reliably estimated by variational methods like MINE (Belghazi et al., 2018). Song & Ermon (2019) further categorized the state-of-the-art variational methods into "generative" and "discriminative" approaches, depending on whether they estimate the probability densities or the density ratios. They showed that the "generative" approaches perform poorly

when the underlying MI is small and "discriminative" approaches perform poorly when MI is large; moreover, certain approaches like MINE (Belghazi et al., 2018) are prone to high variances.

We propose a simple discriminative approach that avoids the limitations of previous discriminative methods that are based on variational bounds. Instead of estimating density or attempting to predict one data variable from another, our method estimates the likelihood that a sample is drawn from the joint distribution versus the product of marginal distributions. A similar classifier-based approach was used by Lopez-Paz & Oquab (2017) for "two sample testing" – hypothesis tests about whether two samples are from the same distribution or not. If the two distributions are the joint and product of the marginals, then the test is for independence. A generalization of this work was used by Sen et al. (2017) to test for conditional independence. We show that accurate performance on this classification task provides an estimate of the log odds. This can greatly simplify the MI estimation task in comparison with generative approaches: estimating a single likelihood ratio may be easier than estimating three distributions (the joint and the two marginals). Moreover, classification tasks are generally amicable to deep learning, while density estimation remains challenging in many cases. Our approach avoids the estimation of the partition function, which induces large variance in most discriminative methods (Song & Ermon, 2019). Our empirical results bear out these conceptual advantages.

Our approach, as well as other sampling-based methods such as MINE, uses the given joint/paired data with derived "unpaired" data that captures the product of the marginal distributions $p(x)p(y)$. The unpaired data can be synthesized via permutations or resampling of the paired data. This construction, which synthesizes unpaired data and then defines a metric to encourage paired data points to map closer than the unpaired data in the latent space, has previously been used in other machine learning applications, such as audio-video and image-text joint representation learning (Harwath et al., 2016; Chauhan et al., 2020). Recent contrastive learning approaches (Tian et al., 2019; Hénaff et al., 2019; Chen et al., 2020; He et al., 2020) further leverage a machine learning model to differentiate paired and unpaired data mostly in the context of unsupervised representation learning. Simonovsky et al. (2016) used paired and unpaired data in conjunction with a classifier-based loss function for patch-based image registration.

This paper is organized as follows. In Section 2, we derive our approach to estimating MI. Section 2.4 discusses connections to related approaches, including MINE. This is followed by empirical evaluation in Section 3. Our experimental results on synthetic and real image data demonstrate the advantages of the proposed discriminative classification-based MI estimator, which has higher accuracy than the state-of-the-art variational approaches and a good bias/variance tradeoff.

## 2    METHODS

Let $x \in \mathcal{X}$ and $y \in \mathcal{Y}$ be two random variables generated by joint distribution $p : \mathcal{X} \times \mathcal{Y} \to \mathbb{R}^+$. Mutual Information (MI)

$$I(x; y) \triangleq \mathbb{E}_{p(x,y)} \left[ \log \frac{p(x, y)}{p(x)p(y)} \right] \tag{1}$$

is a measure of dependence between $x$ and $y$. Let $\mathcal{D} = \{(x_i, y_i)_{i=1}^n\}$ be a set of $n$ independent identically distributed (i.i.d.) samples from $p(x, y)$. The law of large numbers implies

$$\hat{I}_p(\mathcal{D}) \triangleq \frac{1}{n} \sum_{i=1}^n \log \frac{p(x_i, y_i)}{p(x_i)p(y_i)} \to I(x; y) \quad \text{as } n \to \infty, \tag{2}$$

which suggests a simple estimation strategy via sampling. Unfortunately, the joint distribution $p(x, y)$ is often unknown and therefore the estimate in Eq. (2) cannot be explicitly computed. Here we develop an approach to accurately approximating the estimate $\hat{I}_p(\mathcal{D})$ based on discriminative learning.

In our development, we will find it convenient to define a Bernoulli random variable $z \in \{0, 1\}$ and to "lift" the distribution $p(x, y)$ to the product space $\mathcal{X} \times \mathcal{Y} \times \{0, 1\}$. We thus define a family of

distributions parametrized by $\alpha \in (0, 1)$ as follows:

$$p_*(x, y | z = 1; \alpha) = p(x, y), \tag{3}$$
$$p_*(x, y | z = 0; \alpha) = p(x)p(y), \tag{4}$$
$$p_*(z = 1; \alpha) = 1 - p_*(z = 0; \alpha) = \alpha. \tag{5}$$

Using Bayes' rule, we obtain

$$\frac{p_*(z = 1 | x, y)}{p_*(z = 0 | x, y)} = \frac{p_*(x, y, z = 1)}{p_*(x, y, z = 0)} = \frac{p_*(x, y | z = 1) \, p_*(z = 1)}{p_*(x, y | z = 0) \, p_*(z = 0)} = \frac{p(x, y)}{p(x)p(y)} \cdot \frac{\alpha}{1 - \alpha}, \tag{6}$$

which implies that the estimate in (2) can be alternatively expressed as

$$\hat{I}_p = \frac{1}{n} \sum_{i=1}^{n} \log \frac{p_*(z = 1 | x_i, y_i)}{p_*(z = 0 | x_i, y_i)} - \log \frac{\alpha}{1 - \alpha} \tag{7}$$

$$= \frac{1}{n} \sum_{i=1}^{n} \text{logit} \left[ p_*(z = 1 | x_i, y_i) \right] - \text{logit}[\alpha], \tag{8}$$

where $\text{logit}[u] \triangleq \log \frac{u}{1-u}$ is the log-odds function.

Our key idea is to approximate the latent posterior distribution $p_*(z = 1 | x, y)$ by a classifier that is trained to distinguish between the joint distribution $p(x, y)$ and the product distribution $p(x)p(y)$ as described below.

## 2.1 TRAINING SET CONSTRUCTION

We assume that we have access to a large collection $\hat{\mathcal{D}}$ of i.i.d. samples $(x, y)$ from $p(x, y)$ and define $\hat{p}(x, y; \hat{\mathcal{D}})$, $\hat{p}(x; \hat{\mathcal{D}})$, and $\hat{p}(y; \hat{\mathcal{D}})$ to be the empirical joint and marginal distributions respectively induced by data set $\hat{\mathcal{D}}$.

We construct the training set $\mathcal{T} = \{(x^j, y^j, z^j)\}$ of $m$ i.i.d. samples from our empirical approximation to the distribution $p_*(x, y, z)$. Each sample is generated independently of all others as follows. First, a value $z^j \in \{0, 1\}$ is sampled from the prior distribution $p_*(z)$ in (5). If $z^j = 1$, then a pair $(x^j, y^j)$ is sampled randomly from the empirical joint distribution $\hat{p}(x, y; \hat{\mathcal{D}})$; otherwise value $x^j$ is sampled randomly from the empirical marginal distribution $\hat{p}(x; \hat{\mathcal{D}})$ and value $y^j$ is sampled randomly from the empirical marginal distribution $\hat{p}(y; \hat{\mathcal{D}})$, independently from $x^j$. This sampling is easy to implement as it simply samples an element from a set of unique values in the original collection $\hat{\mathcal{D}}$ with frequencies adjusted to account for repeated appearances of the same value.

It is straightforward to verify that any individual sample in the training set $\mathcal{T}$ is generated from distribution $p_*(x, y, z)$ up to the sampling of $\hat{\mathcal{D}}$. Where $\hat{\mathcal{D}}$ is small, multiple samples may not be jointly from $\hat{\mathcal{D}}$ but from some idiosyncratic subset; however, the empirical distribution induced by the set $\mathcal{T}$ converges to $p_*(x, y, z)$ as the size of available data $\hat{\mathcal{D}}$ and the size $m$ of the training set $\mathcal{T}$ becomes large.

## 2.2 CLASSIFIER TRAINING FOR MUTUAL INFORMATION ESTIMATION

Let $q(z = 1 | x, y; \theta, \mathcal{T})$ be a (binary) classifier parameterized by $\theta$ and derived from the training set $\mathcal{T}$. If $q(z = 1 | x, y; \theta, \mathcal{T})$ accurately approximates the posterior distribution $p_*(z = 1 | x, y; \alpha)$, then we can use this classifier $q$ instead of $p_*(z = 1 | x, y; \alpha)$ in (8) to estimate MI.

We follow the widely used maximum likelihood approach to estimating the classifier's parameters $\theta$ and form the cross-entropy loss function

$$\ell(\theta; \mathcal{T}) = -\frac{1}{m} \sum_{j=1}^{m} \log q(z^j | x^j, y^j; \theta, \mathcal{T}) \tag{9}$$

$$= -\frac{1}{m} \sum_{j=1}^{m} z^j \log q(z^j = 1 | x^j, y^j; \theta, \mathcal{T}) + (1 - z^j) \log(1 - q(z^j = 1 | x^j, y^j; \theta, \mathcal{T})) \tag{10}$$

to be minimized to determine the optimal value of parameters $\hat{\theta}$. Once the optimization is completed, we form the estimate

$$\hat{I}_q(\mathcal{D}, \mathcal{T}) = \frac{1}{n} \sum_{i=1}^{n} \text{logit} \left[ q(z = 1 | x_i, y_i; \hat{\theta}, \mathcal{T}) \right] - \text{logit}[\alpha] \tag{11}$$

that approximates the estimate in (8). Note that the estimate is computed using the data set $\mathcal{D}$, which is distinct from the training set $\mathcal{T}$.

## 2.3 ASYMPTOTIC ANALYSIS

As the size of available data $\hat{\mathcal{D}}$ and the size $m$ of the training set $\mathcal{T}$ increase to infinity, the law of large numbers implies

$$\ell(\theta; \mathcal{T}) \to \mathbb{E}_{p_*(x,y,z)} \left[ \log q(z | x, y; \theta, \mathcal{T}) \right], \tag{12}$$

and therefore

$$\hat{\theta} \stackrel{\Delta}{=} \arg \min_{\theta} \ell(\theta; \mathcal{T}) \to \arg \max_{\theta} \mathbb{E}_{p_*(x,y,z)} \left[ \log q(z | x, y; \theta, \mathcal{T}) \right]. \tag{13}$$

Thus, when the model capacity of the family $q(z | x, y; \theta)$ is large enough to include the original distribution $p_*(z | x, y)$, Gibb's inequality implies

$$q(z | x, y; \hat{\theta}, \mathcal{T}) \to p_*(z | x, y) \quad \text{and} \quad \hat{I}_q(\mathcal{D}, \mathcal{T}) \to I(x; y) \tag{14}$$

as both the training data and testing data grow.

## 2.4 CONNECTIONS TO OTHER MUTUAL INFORMATION ESTIMATORS

**MINE and SMILE**  Belghazi et al. (2018) introduced the Mutual Information Neural Estimation (MINE) method, wherein they proposed learning a neural network $f(x, y; \theta)$ that maximizes the objective function $J(f) = \mathbb{E}_{p(x,y)} \left[ f(x, y; \theta) \right] - \log \mathbb{E}_{p(x)p(y)} \left[ e^{f(x,y;\theta)} \right]$, which is the Donsker-Varadhan (DV) lower bound for the Kullback–Leibler (KL) divergence. For analysis purposes, we define $\hat{q}(x, y; \theta) \stackrel{\Delta}{=} \frac{1}{Z} e^{f(x,y;\theta)} p(x)p(y)$, where $Z = \mathbb{E}_{p(x)p(y)} \left[ e^{f(x,y;\theta)} \right]$. By substituting into the definition of $J(\cdot)$ and invoking Gibb's inequality, we obtain

$$J(f) = \mathbb{E}_{p(x,y)} \left[ \log \hat{q}(x, y; \theta) \right] - \mathbb{E}_{p(x,y)} \left[ \log p(x)p(y) \right] \tag{15}$$

$$\leq \mathbb{E}_{p(x,y)} \left[ \log p(x, y) \right] - \mathbb{E}_{p(x,y)} \left[ \log p(x)p(y) \right] = I(x; y), \tag{16}$$

with equality if and only if $\hat{q}(x, y; \theta) \equiv p(x, y)$, i.e.,

$$f(x, y) = \log \frac{p(x, y)}{p(x)p(y)} + C, \tag{17}$$

where C is a constant that is absorbed into the partition function $Z$. Thus the objective function is a lower bound on MI and is maximized when the unspecified "statistics network" $f(x, y)$ is the log likelihood ratio of the joint distribution and the product of the marginals.

Song & Ermon (2019) introduced the Smoothed Mutual Information Lower Bound Estimator (SMILE) approach which is a modification of the MINE estimator. To alleviate the high variance of $f(x, y)$ in practice, the tilting factor $e^{f(x,y)}$ is constrained to the interval $[e^{-\tau}, e^{\tau}]$, for a tuned hyper-parameter $\tau$. As $\tau \to \infty$, SMILE estimates converge to those produced by MINE.

The log likelihood ratio of the joint versus the marginals, which the $f(x, y)$ network from both these methods approximates, is the optimal classifier function for the task defined on our training set $\mathcal{T}$ above. Our parameterization of this ratio makes use of a classifier and the logit transformation. While analytically equivalent, the MINE and SMILE optimization procedures must instead search over ratio functions directly, optimizing $f(x, y) \approx p(x, y)/p(x)p(y)$ itself. Our experimental results demonstrate the advantage of using our estimator in (11).

**CPC**   Oord et al. (2018) proposed a contrastive predictive coding (CPC) method that also maximizes a lower bound

$$J(f) = \mathbb{E}_{p(x,y)} \left[ \frac{1}{N} \sum_{i=1}^{N} \log \frac{f(x_i, y_i; \theta)}{\frac{1}{N} \sum_{j=1}^{N} f(x_i, y_j; \theta)} \right] + \log N \leq I(x; y), \tag{18}$$

where $f(x, y; \theta)$ is a neural network and $N$ is the batch size. CPC is not capable of estimating high underlying MI accurately– it is constrained by their batch size $N$, and this constraint scales logarithmically. In our approach, we do not estimate the likelihood ratio directly, instead we construct an auxiliary variable and "lift" the joint distribution, where we leverage the power of a discriminative neural network classifier. The logit transformation of our classifier response is used to approximate the log likelihood ratio in Eq. (1).

**CCMI**   Mukherjee et al. (2020) recently proposed a classifier based (conditional) MI estimator (CCMI). The classifier $g(x, y; \theta)$ is trained on paired and unpaired sample pairs to yield the posterior probability that the joint distribution $p(x, y)$ (rather than the product of marginals $p(x)p(y)$) generated a sample pair $(x, y)$. Unlike DEMI, the CCMI estimator

$$\hat{I}(x, y) = \mathbb{E}_{p(x,y)} \left[ \text{logit} \left[ g(x, y) \right] \right] - \log \mathbb{E}_{p(x)p(y)} \left[ \frac{1 - g(x, y)}{g(x, y)} \right] \tag{19}$$

still relies on a variational lower bound in Belghazi et al. (2018). The first term above employs paired sample pairs and is identical to our estimator in Eq. (11) for $g(x, y) \triangleq q(z = 1|x, y; \hat{\theta}, \mathcal{T})$ and $\alpha = 0.5$. The second term depends on the unpaired samples and is asymptotically zero. Thus CCMI and DEMI are asymptotically equivalent, but for finite sample sizes CCMI is prone to higher error than DEMI, as we demonstrate empirically later in the paper.

## 3   EXPERIMENTS

We employ two setups widely used in prior work (Song & Ermon, 2019; Belghazi et al., 2018; Poole et al., 2019; Hjelm et al., 2018) to evaluate the proposed estimator and to compare it to the state of the art approaches for estimating MI. In particular, we directly evaluate the accuracy of the resulting estimate in synthetic examples where the true value of MI can be analytically derived and also compare the methods' performance in a representation learning task where the goal is to maximize MI.

Additional experiments that investigate self-consistency and long-run training behavior are reported in Appendices A and B respectively.

### 3.1   MI ESTIMATION

We sample jointly Gaussian variables $x$ and $y$ with known correlation and thus known MI values, which enables us to measure the accuracy of MI estimators when trained on this data. We vary the dimensionality of $x$ and $y$ (20-d, 50-d, and 100-d), the underlying true MI, and the size of the training set (32K, 80K, and 160K) in order to characterize the relative behaviors of different MI estimators. In an additional experiment, we employ an element-wise cubic transformation ($y_i \mapsto y_i^3$) to generate non-linear dependencies in the data. Since deterministic transformations of $x$ and $y$ preserve MI, we can still access ground truth values of MI in this setup. We generate a different set of 10240 samples held out for testing/estimating MI given each training set. We generate 10 independently drawn training and test sets for each two correlated Gaussian variables.

We assess the following estimators in this experiment:

- **DEMI**, the proposed method, with three settings of the parameter $\alpha \in \{0.25, 0.5, 0.75\}$ in Eq. (5).
- **SMILE** (Song & Ermon, 2019), with three settings of the clipping parameter $\tau \in \{1.0, 5.0, \infty\}$. The $\tau = \infty$ case (i.e., no clipping) is equivalent to the **MINE** (Belghazi et al., 2018) objective.
- **InfoNCE** (Oord et al., 2018), the method used for contrastive predictive coding (CPC).

- **CCMI** (Mukherjee et al., 2020).
- A generative model (**GM**), i.e., directly approximating $\log p(x, y)$ and marginals $\log p(x)$ and $\log p(y)$ using a flow network. We note that it is difficult to make comparable parameterizations between **GM**-flow networks and the rest of the methods, and that additionally because the "base" flow distribution is a Gaussian, these networks have a structural advantage for our synthetic tests. They are, in a sense, correctly specified for the Gaussian case, which probably would not happen in real data.

Each estimator uses the same neural network architecture: a multi-layer perceptron with an initial concatenation layer for the $x$ and $y$ inputs, then two fully connected layers with ReLU activations, then a single output. This final layer uses a linear output for **MINE**, **SMILE**, **InfoNCE**, and **CCMI**, and a logistic output for **DEMI**. We use 256 hidden units for each of the fully connected layers. For the **GM** flow network we use the RealNVP scheme (Dinh et al., 2016), which includes a "transformation block" of two 256-unit fully connected layers, each with ReLU activations. This network outputs two parameters, a scale and a shift, both element-wise. This transformation block is repeated three times.

We train each MI estimator for 20 epochs and with the mini-batch size of 64. We employ the Adam optimizer with learning rate parameter 0.0005. The architecture choices above and the optimization settings are comparable with Song & Ermon (2019).

**Results.** Figure 1 reports the MI estimation error ($I(x, y) - \hat{I}(x, y)$) versus the true underlying MI for the experiments with joint Gaussians and joint Gaussians with a cubic transformation, when the size of the training data size is 160K. Appendix C reports additional experimental results from 80K and 32K training samples. The results of **DEMI** with three different settings of $\alpha$ are very close. In Figure 1, we only show **DEMI** ($\alpha = 0.5$). In Appendix C, we report the other two settings as well.

For all experiments, **InfoNCE** substantially underestimated MI. This is due to its log-batchsize ($\log N$) maximum, which saturates quickly relative to the actual mutual information in these regimes. The limited training data setup leads to increased errors of **CCMI** shown in Appendix C, which is consistent with our analysis in Section 2.4.

Overall, for Gaussian variables, the **GM** method performed very well. This is somewhat expected, as its base distribution for the flow network is itself a Gaussian. This trend begins to fall off at higher MI values for the 100-d case. For the cubic case, however, the **GM** method performs quite poorly, perhaps due to the increased model flexibility required for the transformed distribution.

For 20-d Gaussian variables, **MINE** and **SMILE** with both parameter settings overestimated MI in comparison to **DEMI**, which provided estimates that were fairly close to the ground truth values. Appendix B further investigates this behavior. For the 50-d joint Gaussian case, **DEMI** again produced accurate estimates of MI, while **MINE** and **SMILE** underestimated MI substantially. For the 100-d joint Gaussian case, all approaches underestimated MI, with **DEMI** and **CCMI** performing best.

For 20-d joint Gaussians with a cubic transformation, all approaches underestimated MI, **SMILE** ($\tau$ = 5) and **DEMI** performed best. For the 50-d and 100-d cases, all approaches understimated MI, with **DEMI** performing the best.

In summary, **DEMI** performed best or very similar to the best baseline in all the experiments. It further was not sensitive to the setting of its parameter $\alpha$. Its performance relative to the other MI estimators held up with the training data size decreased.

## 3.2 REPRESENTATION LEARNING

Our second experiment demonstrates the viability of **DEMI** as the differentiable loss estimate in a representation learning task. Specifically, we train an encoder on CIFAR10 and CIFAR100 (Krizhevsky et al., 2009) data sets using the Deep InfoMax (Hjelm et al., 2018) criterion. Deep InfoMax learns representations by maximizing mutual information between local features of an input and the output of an encoder, and by matching the representations to a prior distribution. To evaluate the effectiveness of different MI estimators, we only include the MI maximization as the representation learning objective, without prior matching. We compare **DEMI** with **MINE**,

**SMILE**, **InfoNCE**, and **JSD** (Hjelm et al., 2018) for MI estimation and maximization as required by Deep InfoMax.

As discussed in Hjelm et al. (2018), evaluation of the quality of a representation is case-driven and relies on various proxies. We use *classification* as a proxy to evaluate the representations, i.e., we use Deep InfoMax to train an encoder and learn representations from a data set without class labels, and then we freeze the weights of the encoder and train a small fully-connected neural network classifier using the representation as input. We use the *classification* accuracy as a performance proxy to the representation learning and thus the MI estimators. We build two separate classifiers on the last convolutional layer (conv(256,4,4)) and the following fully connected layer (fc(1024)) for *classification* evaluation of the representations, similar to the setup in Hjelm et al. (2018). The size of the input images is $32 \times 32$ and the encoder has the same architecture as the one in Hjelm et al. (2018).

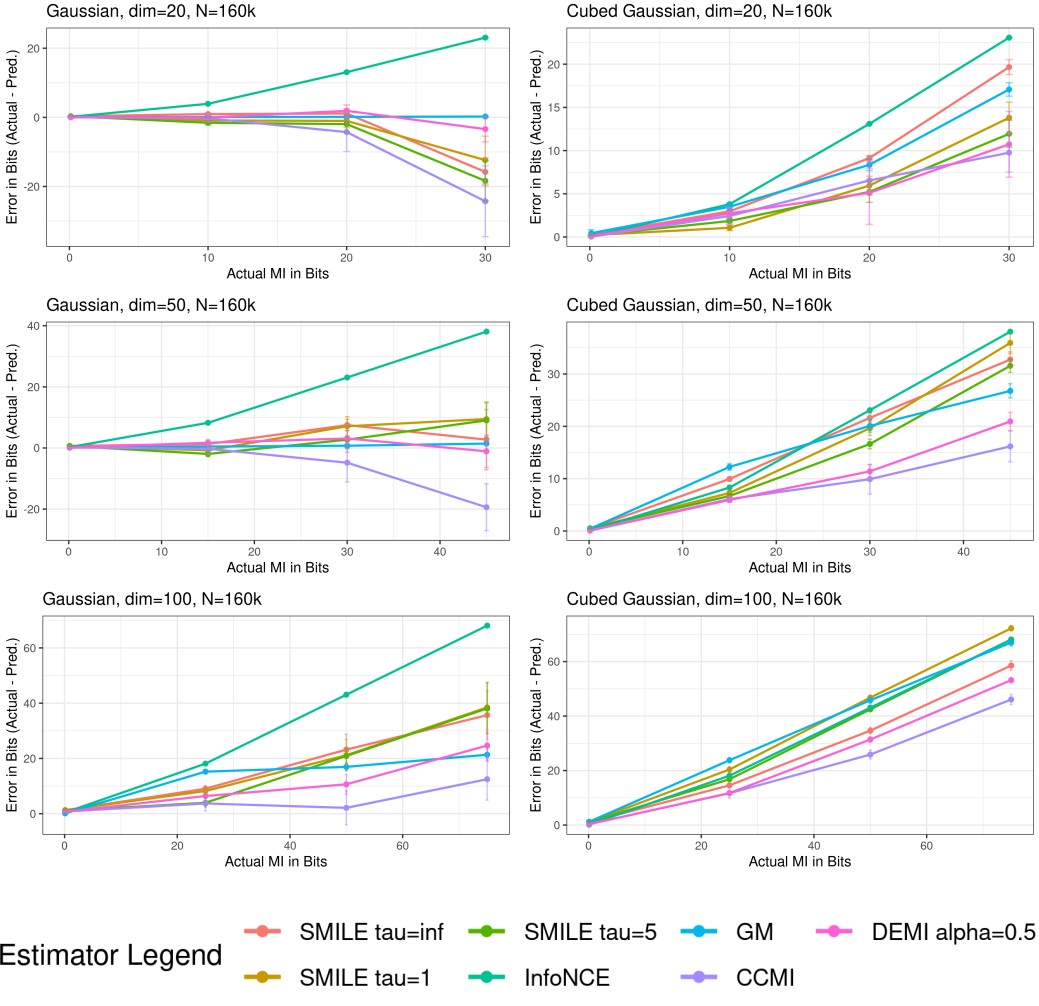

Figure 1: Mutual information estimation between multivariate Gaussian variables (**left column**) and between multivariate Gaussian variables with a cubic transformation (**right column**). **Closer to Zero is better**. The estimation error $(I(x,y) - \hat{I}(x,y))$ versus the true underlying MI are reported. These estimates are based on training data size of 160K. We only show **DEMI** ($\alpha = 0.5$) since the results of the other two parameter settings are very close. A complete table of estimate results is reported in Appendix C

**Results.** Table 1 reports the top 1 classification accuracy of CIFAR10 and CIFAR100. **DEMI** is comparable to **InfoNCE** in 3 out of 4 tasks and outperforms the other MI estimators by a significant margin. When the encoder is allowed to train with the class labels, it becomes a fully-supervised task. We also report its classification accuracy as reference. The classification accuracy based on the representations learned by Deep InfoMax with **DEMI** is close to or even surpasses the fully-supervised case. Note that the Deep InfoMax objective we use here does not include prior distribution matching to regularize the encoder.

Table 1: Top 1 *classification* accuracy as a proxy to representation learning (Deep InfoMax without prior matching) performance with different MI estimators.

|  | CIFAR10 | | CIFAR100 | |
|---|---|---|---|---|
|  | conv(256,4,4) | fc(1024) | conv(256,4,4) | fc(1024) |
| Fully-supervised | 75.39 | | 42.27 | |
| **MINE** | 71.36 | 66.30 | 42.52 | 37.23 |
| **SMILE** ($\tau = 5.0$) | 71.88 | 66.92 | 42.74 | 37.48 |
| **SMILE** ($\tau = 1.0$) | 71.12 | 66.22 | 42.13 | 37.10 |
| **JSD** | 72.79 | 67.94 | 42.78 | 37.76 |
| **InfoNCE** | 73.73 | 69.77 | 44.91 | 39.95 |
| **DEMI** | **74.09** | **70.16** | **45.59** | **42.02** |

## 4 CONCLUSION

We described a simple approach for estimating MI from joint data that is based on a neural network classifier that is trained to distinguish whether a sample pair is drawn from the joint distribution or the product of its marginals. The resulting estimator is the average over joint data of the logit transform of the classifier responses. Theoretically, the estimator converges to MI when the data sizes grow to infinity and the neural network capacity is large enough to contain the corresponding true conditional probability.

The accuracy of our estimator is governed by the ability of the classifier to predict the true posterior probability of items in the test set, which in turn depends on (i) the number of training sample pairs and (ii) the capacity of the neural network used in training. Thus the quality of our estimates is subject to the classical issues of model capacity and overfitting in deep learning. We leave theoretical analysis (which is closely related to the classifier's convergence to the true separating boundary for a general hypothesis class) for future work

We discussed close connections between our approach and the lower bound approaches of MINE and SMILE and InfoNCE(CPC). Unlike the difference-of-entropies (DoE) estimator described in (McAllester & Stratos, 2020), our approach does not make use of assumed distributions. We also demonstrate empirical advantages of our approach over the state of the art methods for estimating MI in synthetic and real image data. Given its simplicity and promising performance, we believe that DEMI is a good candidate for use in research that optimizes MI for representation learning.

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

# A    SELF-CONSISTENCY TESTS

We assess and compare the MI estimators using the self-consistency tests proposed in (Song & Ermon, 2019). We perform the tests on MNIST images (LeCun et al., 2010). The self-consistency tests examine some important properties that a "useful" MI estimator should have, because optimizing MI is more important for many downstream machine learning applications than estimating the exact value of MI.

The self-consistency tests examine: 1) capability of detecting independence, 2) monotonicity with data processing, 3) and additivity. We thus perform the following experiments, where the MNIST image set induces a data distribution and each MNIST image is a random variable that follows this data distribution:

- **MI estimation between one MNIST image and one row-masked image.** Given an MNIST image $X$, we mask out the bottom rows and leave the top $t$ rows of the image, which creates $Y = h(X;t)$. The estimated MI $\hat{I}(X, Y)$ should be equal or very close to zero, if $X$ and $Y$ are independent. In this context, $\hat{I}(X, Y)$ should be close to 0 when $t$ is small and be non-decreasing with $t$. We normalize this measurement to the final value at $t = 28$ (the last row), which should be the maximum information.

- **MI estimation between two identical MNIST images and two row-masked images.** Given an MNIST image $X$, we create two row-masked images: $Y_1 = h(X;t_1)$ and $Y_2 = h(X;t_2)$, where $t_1 = t_2 + 3$. Since additional data processing should not increase mutual information, $\hat{I}([X, X], [Y_1, Y_2])/\hat{I}(X, Y_1)$ should be close to 1.

- **MI estimation between two MNIST images and two row-masked images.** We randomly select two MNIST images and concatenate them: $[X_1, X_2]$, and mask the same number of rows on them: $[h(X_1;t), h(X_2;t)] = [Y_1, Y_2]$. $\hat{I}([X_1, X_2], [Y_1, Y_2])/\hat{I}(X_1, Y_1)$ should be close to 2.

We have 60k MNIST images or concatnated images for training and a test set of 10k images. We train each MI estimator for 100 epochs and set the mini-batch size to 64. For all methods, we concatentate inputs, then convolve with a $5 \times 5$ kernel with stride 2 and 64 output channels, then apply a fully connected layer with 1024 hidden units, which then maps to a single output. ReLU is applied after all but the last layer.

**Results.**    In Figure 2 we plot the results of the three self-consistency metrics for each method. In general most methods perform well for the first measurement (monotonicity) with the exception of SMILE ($\tau = \infty$), which exhibits charateristicly high variance. Other settings of SMILE and DEMI both are relatively well behaved, though overall InfoNCE performs best. For the second metric ("data processing"), all methods perform well, again aside from SMILE ($\tau = \infty$) variance. In the third metric, SMILE $\tau = 1$ also exhibits a large bump in the center (where optimal should be constant 2 overall), but both InfoNCE and DEMI converge to 1 overall, and no method performs optimally. In general InfoNCE performs best across between the first two measures, but DEMI and SMILE ($\tau = 1$) also do well in two of three.

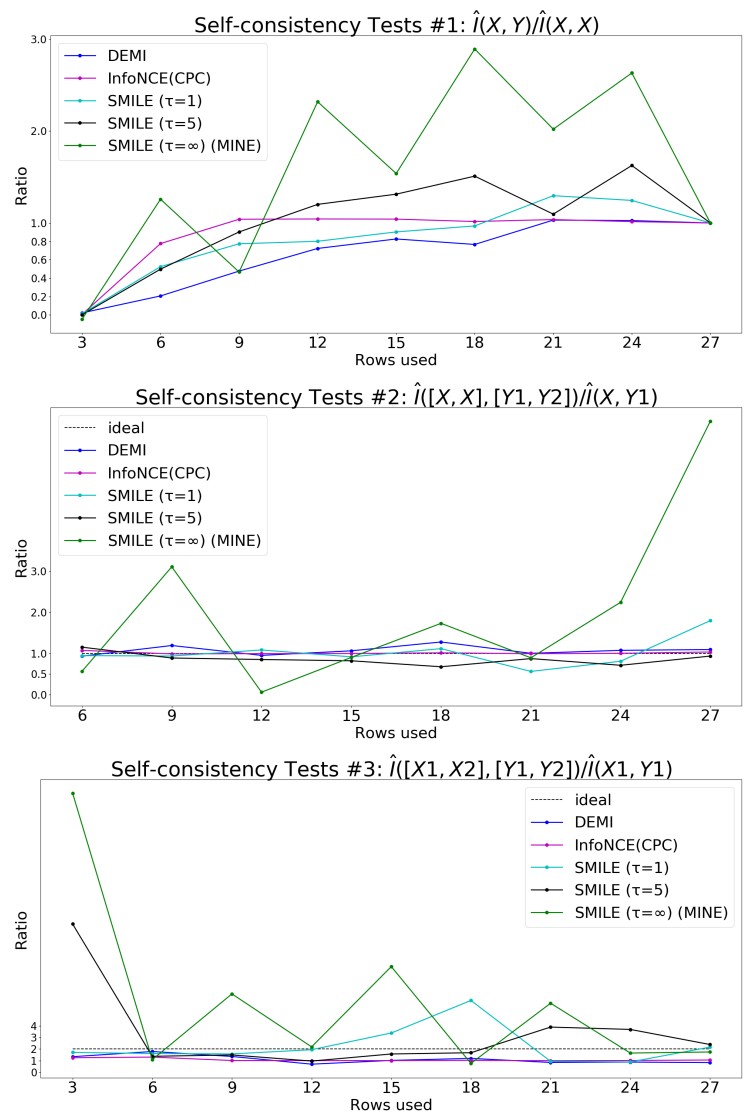

Figure 2: Results of the three self-consistency tests for SMILE ($\tau = 1, 5, \infty$), InfoNCE, and DEMI. "Monotonicity" is **top**, "data processing" is **middle**, and "additivity" is **bottom**.

# B  LONG-RUN TRAINING BEHAVIOR OF SMILE

As shown in Section 3, SMILE somewhat overestimates the MI for the 20 dimensional Gaussian case in high MI regimes ($\sim 30$ Nats or more). This did not occur at lower MI conditions or in higher dimensions.

Further investigation showed this problem to increase as training went on; to illustrate this, we set up a new experiment on the 20 dimensional Gaussian case. We ran each setting of SMILE ($\tau = 1, 5, \infty$) for 100000 training steps with batch size 64, drawing samples directly from the generating distributions. This means that the training set has effectively a very large size. We did this for three ground-truth MI values of 10,20, and 30. For comparison we also run the proposed method through the same.

This setup exactly mirrors the experiment in (Song & Ermon, 2019) Figure 1 in Section 6.1 of that paper, and uses their provided code and generation method, except that we replace their step-wise increasing MI schedule with a constant 10, 20, or 30 nat generator, and we run the experiment longer.

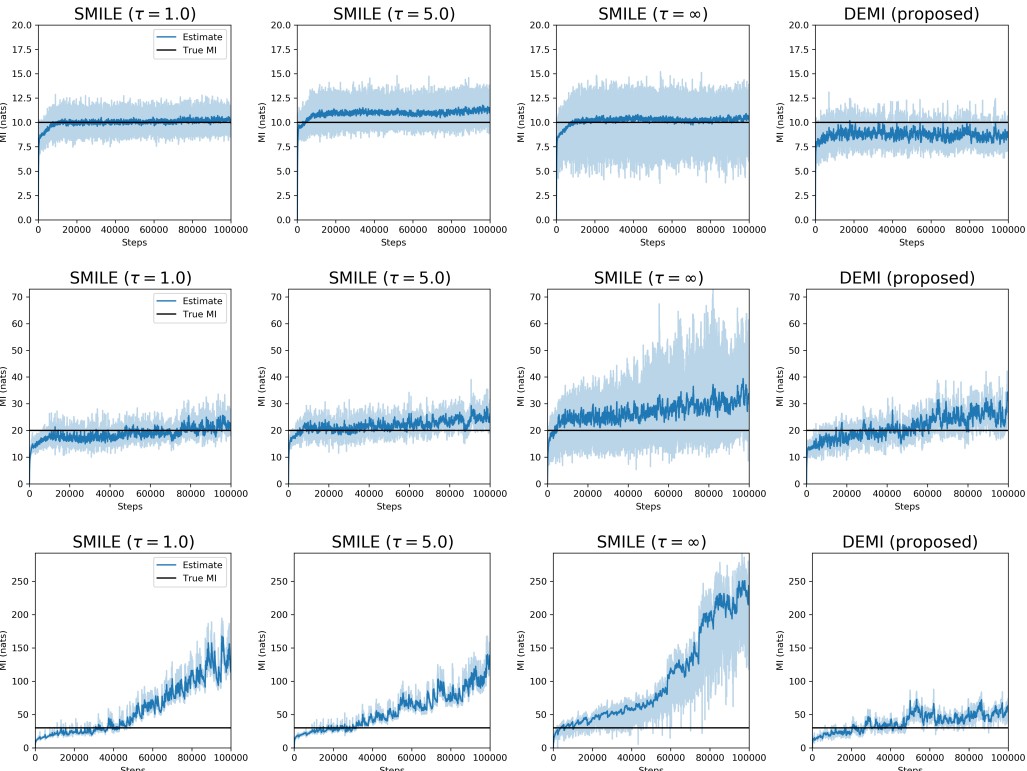

Figure 3: Long-run behavior of SMILE and DEMI for 10 (**top row**), 20 (**middle row**), and 30 (**bottom row**) Nats. Analytically SMILE converges to the MINE objective for $\tau \to \infty$. Smoothed trajectories are plotted in bold, exact trajectories are the semi-translucent curve, and the actual Mutual information is the black constant line.

The curves for the first row of Figure 3 show good performance with relatively stable long-term behavior, particularly for $\tau = 1$. The curves in the third row of Figure 3 on the otherhand suggest that for certain distribution/domain combinations, even though SMILE and MINE are based on a lower bound of MI, they can both grossly overestimate it. This may be as (McAllester & Stratos, 2020) suggests due in part to a sensitivity of the estimate of $-\ln \mathbb{E}[e^{f(x,y)}]$ to outliers. The proposed method eventually overestimates as well in both the 20 and 30 nat cases, but does not have the strongly divergent behavior exhibited by SMILE (seen particularly strongly in $\tau = \infty$ settings).

## C    RESULTS OF MI ESTIMATION ON GAUSSIAN VARIABLES

We report the results in the tables below of MI estimation on Gaussian variables described in Section 3.1. The estimation error $(I(x, y) - \hat{I}(x, y))$ and the standard deviation of the estimates are reported.

Table 2: MI estimation between 20-d Gaussian variables trained on 160K data samples. The estimation error $(I(x, y) - \hat{I}(x, y))$ and the standard deviation of the estimates are reported.

| Actual MI | SMILE $\tau = \infty$ | SMILE $\tau = 1$ | SMILE $\tau = 5$ | InfoNCE | GM |
|---|---|---|---|---|---|
| 0.1 | $0.3 \pm 0.0$ | $0.3 \pm 0.0$ | $0.3 \pm 0.0$ | $0.2 \pm 0.0$ | $0.1 \pm 0.0$ |
| 5 | $0.7 \pm 0.1$ | $-0.2 \pm 0.3$ | $0.2 \pm 0.1$ | $1.0 \pm 0.1$ | $0.1 \pm 0.0$ |
| 10 | $0.9 \pm 0.2$ | $-0.9 \pm 0.6$ | $-1.6 \pm 0.6$ | $3.9 \pm 0.0$ | $0.1 \pm 0.0$ |
| 15 | $0.2 \pm 1.0$ | $-2.6 \pm 1.7$ | $-5.2 \pm 1.8$ | $8.1 \pm 0.0$ | $0.1 \pm 0.0$ |
| 20 | $1.1 \pm 1.4$ | $-1.0 \pm 2.0$ | $-2.0 \pm 2.4$ | $13.1 \pm 0.0$ | $0.1 \pm 0.1$ |
| 25 | $-6.1 \pm 4.6$ | $-12.6 \pm 8.7$ | $-15.3 \pm 13.1$ | $18.1 \pm 0.0$ | $0.2 \pm 0.0$ |
| 30 | $-15.8 \pm 4.0$ | $-12.4 \pm 6.9$ | $-18.4 \pm 5.8$ | $23.1 \pm 0.0$ | $0.2 \pm 0.1$ |
| 35 | $-13.5 \pm 7.1$ | $-4.5 \pm 4.3$ | $-8.5 \pm 6.8$ | $28.1 \pm 0.0$ | $0.3 \pm 0.0$ |

| Actual MI | CCMI | DEMI $\alpha = 0.5$ | DEMI $\alpha = 0.25$ | DEMI $\alpha = 0.75$ |
|---|---|---|---|---|
| 0.1 | $0.0 \pm 0.0$ | $0.0 \pm 0.0$ | $-0.6 \pm 0.0$ | $0.5 \pm 0.0$ |
| 5 | $0.4 \pm 0.6$ | $0.0 \pm 0.1$ | $-0.6 \pm 0.2$ | $0.5 \pm 0.2$ |
| 10 | $-0.3 \pm 1.1$ | $-0.0 \pm 0.6$ | $-0.9 \pm 0.4$ | $0.9 \pm 0.4$ |
| 15 | $-1.7 \pm 3.1$ | $0.9 \pm 1.1$ | $0.0 \pm 1.1$ | $1.7 \pm 0.8$ |
| 20 | $-4.3 \pm 5.6$ | $1.9 \pm 1.7$ | $0.0 \pm 2.1$ | $1.3 \pm 2.5$ |
| 25 | $-15.6 \pm 6.8$ | $0.3 \pm 3.8$ | $-0.9 \pm 3.9$ | $-3.2 \pm 5.9$ |
| 30 | $-24.3 \pm 10.2$ | $-3.4 \pm 3.7$ | $-0.5 \pm 1.8$ | $-3.4 \pm 4.8$ |
| 35 | $-27.3 \pm 8.0$ | $0.4 \pm 9.1$ | $-1.9 \pm 3.6$ | $-0.3 \pm 3.2$ |

Table 3: MI estimation between 20-d Gaussian variables with cubic transformation trained on 160K data samples. The estimation error $(I(x, y) - \hat{I}(x, y))$ and the standard deviation of the estimates are reported.

| Actual MI | SMILE $\tau = \infty$ | SMILE $\tau = 1$ | SMILE $\tau = 5$ | InfoNCE | GM |
|---|---|---|---|---|---|
| 0.1 | $0.2 \pm 0.0$ | $0.2 \pm 0.0$ | $0.2 \pm 0.0$ | $0.1 \pm 0.0$ | $0.4 \pm 0.4$ |
| 5 | $1.3 \pm 0.1$ | $0.2 \pm 0.1$ | $1.1 \pm 0.1$ | $0.9 \pm 0.0$ | $2.7 \pm 0.2$ |
| 10 | $3.0 \pm 0.1$ | $1.1 \pm 0.3$ | $1.9 \pm 0.2$ | $3.8 \pm 0.0$ | $3.5 \pm 0.4$ |
| 15 | $5.6 \pm 0.4$ | $2.8 \pm 0.6$ | $2.7 \pm 0.7$ | $8.2 \pm 0.0$ | $5.7 \pm 0.4$ |
| 20 | $9.1 \pm 0.3$ | $5.9 \pm 1.1$ | $5.2 \pm 1.2$ | $13.1 \pm 0.0$ | $8.4 \pm 0.7$ |
| 25 | $14.0 \pm 0.6$ | $10.3 \pm 2.1$ | $7.5 \pm 1.0$ | $18.1 \pm 0.0$ | $12.2 \pm 0.7$ |
| 30 | $19.7 \pm 0.9$ | $13.8 \pm 1.8$ | $11.9 \pm 1.6$ | $23.1 \pm 0.0$ | $17.1 \pm 0.8$ |
| 35 | $26.0 \pm 0.9$ | $20.6 \pm 1.5$ | $18.3 \pm 1.6$ | $28.1 \pm 0.0$ | $22.8 \pm 0.9$ |

| Actual MI | CCMI | DEMI $\alpha = 0.5$ | DEMI $\alpha = 0.25$ | DEMI $\alpha = 0.75$ |
|---|---|---|---|---|
| 0.1 | $0.0 \pm 0.0$ | $0.1 \pm 0.0$ | $-0.6 \pm 0.0$ | $0.5 \pm 0.0$ |
| 5 | $0.9 \pm 0.2$ | $1.0 \pm 0.4$ | $0.0 \pm 0.1$ | $1.4 \pm 0.2$ |
| 10 | $2.4 \pm 1.0$ | $2.7 \pm 0.6$ | $1.5 \pm 0.2$ | $2.9 \pm 0.2$ |
| 15 | $4.8 \pm 2.3$ | $4.1 \pm 1.7$ | $3.1 \pm 0.4$ | $4.7 \pm 0.5$ |
| 20 | $6.5 \pm 1.4$ | $5.1 \pm 3.6$ | $5.4 \pm 0.5$ | $6.1 \pm 0.6$ |
| 25 | $8.8 \pm 3.7$ | $7.6 \pm 3.1$ | $8.4 \pm 0.7$ | $9.1 \pm 0.5$ |
| 30 | $9.8 \pm 2.3$ | $10.7 \pm 3.8$ | $12.5 \pm 1.0$ | $13.1 \pm 1.1$ |
| 35 | $16.7 \pm 5.2$ | $13.9 \pm 3.6$ | $16.9 \pm 0.8$ | $17.2 \pm 1.2$ |

Table 4: MI estimation between 20-d Gaussian variables trained on 80K data samples. The estimation error $(I(x,y) - \hat{I}(x,y))$ and the standard deviation of the estimates are reported.

| Actual MI | SMILE $\tau = \infty$ | SMILE $\tau = 1$ | SMILE $\tau = 5$ | InfoNCE | GM |
|---|---|---|---|---|---|
| 0.1 | $0.6 \pm 0.1$ | $0.6 \pm 0.1$ | $0.6 \pm 0.1$ | $0.3 \pm 0.0$ | $0.1 \pm 0.0$ |
| 5 | $1.0 \pm 0.5$ | $-0.1 \pm 0.5$ | $0.5 \pm 0.6$ | $1.6 \pm 0.2$ | $0.1 \pm 0.0$ |
| 10 | $1.5 \pm 1.1$ | $-0.5 \pm 1.3$ | $-1.1 \pm 1.6$ | $4.0 \pm 0.1$ | $0.2 \pm 0.1$ |
| 15 | $1.9 \pm 2.3$ | $-0.1 \pm 1.8$ | $-2.2 \pm 3.3$ | $8.1 \pm 0.0$ | $0.2 \pm 0.0$ |
| 20 | $3.0 \pm 4.3$ | $1.7 \pm 3.5$ | $-2.0 \pm 4.2$ | $13.1 \pm 0.0$ | $0.3 \pm 0.1$ |
| 25 | $0.4 \pm 9.2$ | $-2.9 \pm 10.3$ | $-7.8 \pm 8.8$ | $18.1 \pm 0.0$ | $0.4 \pm 0.1$ |
| 30 | $-5.1 \pm 11.5$ | $-1.3 \pm 7.5$ | $-6.1 \pm 8.7$ | $23.1 \pm 0.0$ | $0.5 \pm 0.1$ |
| 35 | $1.6 \pm 12.9$ | $6.5 \pm 8.9$ | $4.6 \pm 9.5$ | $28.1 \pm 0.0$ | $0.8 \pm 0.1$ |

| Actual MI | CCMI | DEMI $\alpha = 0.5$ | DEMI $\alpha = 0.25$ | DEMI $\alpha = 0.75$ |
|---|---|---|---|---|
| 0.1 | $0.0 \pm 0.0$ | $0.1 \pm 0.0$ | $-0.6 \pm 0.0$ | $0.5 \pm 0.0$ |
| 5 | $0.4 \pm 0.6$ | $0.0 \pm 0.2$ | $-0.6 \pm 0.1$ | $0.6 \pm 0.2$ |
| 10 | $-0.3 \pm 1.1$ | $0.4 \pm 0.4$ | $-0.3 \pm 0.4$ | $1.2 \pm 0.5$ |
| 15 | $-1.7 \pm 3.1$ | $1.3 \pm 1.2$ | $0.0 \pm 0.8$ | $1.6 \pm 1.4$ |
| 20 | $-4.3 \pm 5.6$ | $1.2 \pm 1.7$ | $1.2 \pm 1.8$ | $2.1 \pm 1.8$ |
| 25 | $-15.6 \pm 6.8$ | $2.6 \pm 3.4$ | $1.5 \pm 3.2$ | $1.5 \pm 1.9$ |
| 30 | $-24.3 \pm 10.2$ | $4.3 \pm 2.5$ | $3.4 \pm 2.0$ | $4.7 \pm 1.4$ |
| 35 | $-27.3 \pm 8.0$ | $7.7 \pm 2.3$ | $8.8 \pm 2.0$ | $8.9 \pm 2.4$ |

Table 5: MI estimation between 20-d Gaussian variables with cubic transformation trained on 80K data samples. The estimation error $(I(x,y) - \hat{I}(x,y))$ and the standard deviation of the estimates are reported.

| Actual MI | SMILE $\tau = \infty$ | SMILE $\tau = 1$ | SMILE $\tau = 5$ | InfoNCE | GM |
|---|---|---|---|---|---|
| 0.1 | $0.3 \pm 0.0$ | $0.4 \pm 0.1$ | $0.3 \pm 0.1$ | $0.1 \pm 0.0$ | $0.2 \pm 0.2$ |
| 5 | $1.8 \pm 0.2$ | $0.8 \pm 0.3$ | $1.6 \pm 0.1$ | $1.0 \pm 0.0$ | $3.0 \pm 0.3$ |
| 10 | $3.8 \pm 0.2$ | $2.2 \pm 0.4$ | $2.8 \pm 0.2$ | $3.9 \pm 0.1$ | $4.8 \pm 0.6$ |
| 15 | $6.8 \pm 0.2$ | $4.8 \pm 0.6$ | $4.6 \pm 0.7$ | $8.2 \pm 0.0$ | $6.9 \pm 0.5$ |
| 20 | $11.1 \pm 0.3$ | $8.4 \pm 1.2$ | $7.2 \pm 1.7$ | $13.1 \pm 0.0$ | $9.9 \pm 0.8$ |
| 25 | $16.3 \pm 0.6$ | $12.2 \pm 2.3$ | $11.5 \pm 1.4$ | $18.1 \pm 0.0$ | $13.0 \pm 0.8$ |
| 30 | $22.8 \pm 1.7$ | $18.9 \pm 1.2$ | $16.5 \pm 1.2$ | $23.1 \pm 0.0$ | $16.8 \pm 0.4$ |
| 35 | $28.5 \pm 0.9$ | $24.3 \pm 2.6$ | $22.4 \pm 1.3$ | $28.1 \pm 0.0$ | $21.4 \pm 0.6$ |

| Actual MI | CCMI | DEMI $\alpha = 0.5$ | DEMI $\alpha = 0.25$ | DEMI $\alpha = 0.75$ |
|---|---|---|---|---|
| 0.1 | $0.1 \pm 0.0$ | $0.1 \pm 0.0$ | $-0.6 \pm 0.0$ | $0.5 \pm 0.0$ |
| 5 | $1.4 \pm 0.4$ | $1.0 \pm 0.1$ | $0.3 \pm 0.2$ | $1.7 \pm 0.2$ |
| 10 | $4.2 \pm 1.1$ | $2.5 \pm 0.4$ | $1.8 \pm 0.4$ | $3.3 \pm 0.3$ |
| 15 | $5.6 \pm 1.9$ | $4.8 \pm 0.5$ | $3.7 \pm 0.5$ | $4.9 \pm 0.5$ |
| 20 | $5.9 \pm 2.2$ | $6.6 \pm 0.5$ | $6.3 \pm 0.6$ | $7.6 \pm 0.8$ |
| 25 | $9.4 \pm 2.3$ | $10.2 \pm 0.7$ | $10.3 \pm 0.7$ | $10.8 \pm 1.0$ |
| 30 | $15.7 \pm 5.9$ | $14.4 \pm 1.1$ | $14.0 \pm 0.7$ | $14.6 \pm 0.8$ |
| 35 | $16.3 \pm 2.7$ | $18.9 \pm 0.7$ | $18.0 \pm 0.5$ | $19.5 \pm 1.0$ |

Table 6: MI estimation between 20-d Gaussian variables trained on 32K data samples. The estimation error $(I(x, y) - \hat{I}(x, y))$ and the standard deviation of the estimates are reported.

| Actual MI | SMILE $\tau = \infty$ | SMILE $\tau = 1$ | SMILE $\tau = 5$ | InfoNCE | GM |
|---|---|---|---|---|---|
| 0.1 | $1.1 \pm 0.3$ | $1.2 \pm 0.3$ | $1.2 \pm 0.2$ | $0.8 \pm 0.1$ | $0.1 \pm 0.0$ |
| 5 | $0.9 \pm 0.1$ | $-0.0 \pm 0.5$ | $0.5 \pm 0.1$ | $3.9 \pm 0.3$ | $1.4 \pm 0.2$ |
| 10 | $1.3 \pm 0.2$ | $-0.5 \pm 1.0$ | $-1.0 \pm 0.6$ | $4.0 \pm 0.1$ | $0.8 \pm 0.1$ |
| 15 | $1.6 \pm 1.3$ | $-1.2 \pm 1.3$ | $-2.9 \pm 1.4$ | $8.1 \pm 0.0$ | $0.8 \pm 0.1$ |
| 20 | $-2.3 \pm 1.5$ | $-3.8 \pm 3.7$ | $-6.8 \pm 3.9$ | $13.1 \pm 0.0$ | $1.2 \pm 0.2$ |
| 25 | $-0.1 \pm 1.3$ | $1.1 \pm 0.7$ | $-3.4 \pm 0.9$ | $18.1 \pm 0.0$ | $1.6 \pm 0.4$ |
| 30 | $2.5 \pm 1.1$ | $7.0 \pm 0.5$ | $3.1 \pm 0.7$ | $23.1 \pm 0.0$ | $2.3 \pm 0.4$ |
| 35 | $6.9 \pm 0.7$ | $12.3 \pm 0.7$ | $8.7 \pm 0.4$ | $28.1 \pm 0.0$ | $3.8 \pm 0.8$ |

| Actual MI | CCMI | DEMI $\alpha = 0.5$ | DEMI $\alpha = 0.25$ | DEMI $\alpha = 0.75$ |
|---|---|---|---|---|
| 0.1 | $3.6 \pm 0.7$ | $0.3 \pm 0.1$ | $-0.4 \pm 0.0$ | $0.5 \pm 0.0$ |
| 5 | $49.2 \pm 2.5$ | $0.2 \pm 0.2$ | $-0.5 \pm 0.2$ | $0.7 \pm 0.2$ |
| 10 | $97.7 \pm 6.1$ | $0.7 \pm 0.5$ | $0.1 \pm 0.3$ | $1.2 \pm 0.7$ |
| 15 | $91.6 \pm 4.9$ | $1.1 \pm 1.2$ | $0.5 \pm 1.1$ | $1.2 \pm 1.0$ |
| 20 | $91.3 \pm 6.8$ | $2.3 \pm 1.1$ | $2.1 \pm 0.8$ | $3.7 \pm 0.9$ |
| 25 | $98.7 \pm 6.8$ | $5.4 \pm 1.7$ | $5.6 \pm 1.4$ | $6.1 \pm 0.5$ |
| 30 | $97.8 \pm 6.4$ | $10.0 \pm 1.8$ | $9.9 \pm 1.0$ | $11.5 \pm 2.1$ |
| 35 | $103.2 \pm 4.6$ | $15.8 \pm 1.5$ | $15.5 \pm 1.7$ | $16.5 \pm 2.0$ |

Table 7: MI estimation between 20-d Gaussian variables with cubic transformation trained on 32K data samples. The estimation error $(I(x, y) - \hat{I}(x, y))$ and the standard deviation of the estimates are reported.

| Actual MI | SMILE $\tau = \infty$ | SMILE $\tau = 1$ | SMILE $\tau = 5$ | InfoNCE | GM |
|---|---|---|---|---|---|
| 0.1 | $0.7 \pm 0.1$ | $0.7 \pm 0.2$ | $0.7 \pm 0.1$ | $0.3 \pm 0.0$ | $0.6 \pm 0.6$ |
| 5 | $2.3 \pm 0.1$ | $1.2 \pm 0.3$ | $2.2 \pm 0.1$ | $1.2 \pm 0.0$ | $4.3 \pm 0.4$ |
| 10 | $4.7 \pm 0.2$ | $2.9 \pm 0.5$ | $3.7 \pm 0.3$ | $4.0 \pm 0.1$ | $8.1 \pm 0.4$ |
| 15 | $8.4 \pm 0.4$ | $5.8 \pm 1.2$ | $5.6 \pm 1.4$ | $8.3 \pm 0.0$ | $11.6 \pm 0.7$ |
| 20 | $13.7 \pm 1.0$ | $9.0 \pm 1.1$ | $8.8 \pm 1.0$ | $13.1 \pm 0.0$ | $15.9 \pm 0.9$ |
| 25 | $19.4 \pm 1.3$ | $15.0 \pm 2.0$ | $13.2 \pm 1.1$ | $18.1 \pm 0.0$ | $20.9 \pm 0.9$ |
| 30 | $26.8 \pm 1.5$ | $20.7 \pm 1.6$ | $18.7 \pm 0.8$ | $23.1 \pm 0.0$ | $25.2 \pm 1.3$ |
| 35 | $32.3 \pm 1.3$ | $28.0 \pm 1.8$ | $25.5 \pm 0.8$ | $28.1 \pm 0.0$ | $28.6 \pm 1.3$ |

| Actual MI | CCMI | DEMI $\alpha = 0.5$ | DEMI $\alpha = 0.25$ | DEMI $\alpha = 0.75$ |
|---|---|---|---|---|
| 0.1 | $0.1 \pm 0.0$ | $0.1 \pm 0.0$ | $-0.5 \pm 0.0$ | $0.5 \pm 0.0$ |
| 5 | $1.8 \pm 0.1$ | $1.6 \pm 0.2$ | $0.8 \pm 0.1$ | $2.0 \pm 0.2$ |
| 10 | $4.5 \pm 0.4$ | $3.2 \pm 0.4$ | $2.5 \pm 0.3$ | $3.5 \pm 0.2$ |
| 15 | $7.3 \pm 2.8$ | $5.4 \pm 0.5$ | $4.5 \pm 0.5$ | $5.2 \pm 0.7$ |
| 20 | $8.8 \pm 4.4$ | $7.9 \pm 0.8$ | $7.7 \pm 0.7$ | $8.3 \pm 0.7$ |
| 25 | $12.4 \pm 2.3$ | $12.0 \pm 0.7$ | $11.0 \pm 0.5$ | $13.0 \pm 0.7$ |
| 30 | $15.8 \pm 2.9$ | $16.1 \pm 0.4$ | $15.8 \pm 0.7$ | $16.9 \pm 0.9$ |
| 35 | $21.3 \pm 3.7$ | $21.0 \pm 1.0$ | $20.2 \pm 0.7$ | $21.4 \pm 1.0$ |

Table 8: MI estimation between 50-d Gaussian variables trained on 160K data samples. The estimation error $(I(x, y) - \hat{I}(x, y))$ and the standard deviation of the estimates are reported.

| Actual MI | SMILE $\tau = \infty$ | SMILE $\tau = 1$ | SMILE $\tau = 5$ | InfoNCE | GM |
|---|---|---|---|---|---|
| 0.1 | $0.7 \pm 0.2$ | $0.6 \pm 0.1$ | $0.6 \pm 0.1$ | $0.3 \pm 0.0$ | $0.1 \pm 0.0$ |
| 5 | $1.1 \pm 0.1$ | $0.2 \pm 0.2$ | $0.7 \pm 0.1$ | $1.7 \pm 0.1$ | $3.4 \pm 0.3$ |
| 10 | $1.6 \pm 0.2$ | $0.1 \pm 0.4$ | $-0.4 \pm 0.6$ | $4.3 \pm 0.1$ | $1.0 \pm 0.2$ |
| 15 | $1.3 \pm 0.6$ | $-0.8 \pm 1.5$ | $-1.9 \pm 0.9$ | $8.2 \pm 0.0$ | $0.5 \pm 0.2$ |
| 20 | $2.5 \pm 0.9$ | $2.0 \pm 3.0$ | $-0.9 \pm 2.6$ | $13.1 \pm 0.0$ | $0.5 \pm 0.1$ |
| 25 | $4.8 \pm 1.1$ | $4.3 \pm 3.0$ | $0.7 \pm 2.9$ | $18.1 \pm 0.0$ | $0.6 \pm 0.1$ |
| 30 | $7.5 \pm 1.9$ | $7.1 \pm 3.1$ | $2.7 \pm 3.0$ | $23.1 \pm 0.0$ | $0.7 \pm 0.1$ |
| 35 | $6.8 \pm 3.6$ | $6.9 \pm 3.9$ | $1.9 \pm 3.9$ | $28.1 \pm 0.0$ | $0.9 \pm 0.2$ |
| 40 | $7.4 \pm 3.5$ | $8.5 \pm 1.8$ | $5.1 \pm 3.3$ | $33.1 \pm 0.0$ | $1.1 \pm 0.2$ |
| 45 | $2.7 \pm 9.8$ | $9.5 \pm 5.6$ | $9.0 \pm 5.8$ | $38.1 \pm 0.0$ | $1.4 \pm 0.2$ |
| 50 | $3.5 \pm 7.0$ | $14.3 \pm 5.2$ | $7.5 \pm 3.1$ | $43.1 \pm 0.0$ | $1.6 \pm 0.3$ |
| 55 | $10.2 \pm 4.5$ | $19.2 \pm 4.7$ | $15.3 \pm 5.6$ | $48.1 \pm 0.0$ | $1.8 \pm 0.3$ |

| Actual MI | CCMI | DEMI $\alpha = 0.5$ | DEMI $\alpha = 0.25$ | DEMI $\alpha = 0.75$ |
|---|---|---|---|---|
| 0.1 | $0.2 \pm 0.0$ | $0.1 \pm 0.0$ | $-0.5 \pm 0.0$ | $0.5 \pm 0.0$ |
| 5 | $1.1 \pm 0.8$ | $0.1 \pm 0.2$ | $-0.6 \pm 0.1$ | $0.7 \pm 0.2$ |
| 10 | $0.0 \pm 0.5$ | $0.5 \pm 0.4$ | $-0.7 \pm 0.5$ | $1.3 \pm 0.5$ |
| 15 | $-0.3 \pm 1.8$ | $1.7 \pm 1.0$ | $0.8 \pm 0.8$ | $1.6 \pm 1.4$ |
| 20 | $-3.8 \pm 2.0$ | $3.1 \pm 2.0$ | $2.0 \pm 1.7$ | $2.1 \pm 2.4$ |
| 25 | $-4.9 \pm 2.8$ | $1.8 \pm 1.8$ | $2.3 \pm 2.6$ | $1.7 \pm 3.1$ |
| 30 | $-4.8 \pm 6.3$ | $3.1 \pm 4.5$ | $3.0 \pm 3.4$ | $1.1 \pm 2.3$ |
| 35 | $-8.9 \pm 8.1$ | $-0.3 \pm 5.1$ | $2.0 \pm 4.0$ | $0.4 \pm 4.1$ |
| 40 | $-15.3 \pm 7.8$ | $-0.4 \pm 6.7$ | $2.7 \pm 3.6$ | $-0.2 \pm 4.5$ |
| 45 | $-19.4 \pm 7.7$ | $-1.1 \pm 5.3$ | $3.5 \pm 7.6$ | $-1.0 \pm 3.4$ |
| 50 | $-19.6 \pm 6.7$ | $3.2 \pm 8.3$ | $1.2 \pm 4.5$ | $5.3 \pm 2.8$ |
| 55 | $-18.6 \pm 12.5$ | $4.6 \pm 8.0$ | $6.4 \pm 4.5$ | $7.7 \pm 4.3$ |

Table 9: MI estimation between 50-d Gaussian variables with cubic transformation trained on 160K data samples. The estimation error $(I(x, y) - \hat{I}(x, y))$ and the standard deviation of the estimates are reported.

| Actual MI | SMILE $\tau = \infty$ | SMILE $\tau = 1$ | SMILE $\tau = 5$ | InfoNCE | GM |
|---|---|---|---|---|---|
| 0.1 | $0.5 \pm 0.1$ | $0.5 \pm 0.1$ | $0.5 \pm 0.1$ | $0.1 \pm 0.0$ | $0.4 \pm 0.4$ |
| 5 | $3.1 \pm 0.3$ | $2.5 \pm 0.1$ | $3.0 \pm 0.2$ | $1.3 \pm 0.0$ | $4.9 \pm 0.5$ |
| 10 | $7.0 \pm 0.5$ | $4.6 \pm 0.4$ | $5.3 \pm 0.2$ | $4.2 \pm 0.0$ | $8.7 \pm 0.7$ |
| 15 | $9.9 \pm 0.4$ | $7.3 \pm 0.5$ | $6.7 \pm 0.3$ | $8.3 \pm 0.0$ | $12.2 \pm 0.6$ |
| 20 | $13.3 \pm 0.6$ | $10.9 \pm 0.5$ | $9.2 \pm 0.5$ | $13.1 \pm 0.0$ | $15.5 \pm 0.7$ |
| 25 | $17.4 \pm 0.6$ | $15.2 \pm 1.1$ | $12.6 \pm 0.8$ | $18.1 \pm 0.0$ | $17.4 \pm 0.8$ |
| 30 | $21.6 \pm 0.9$ | $19.6 \pm 1.4$ | $16.6 \pm 0.9$ | $23.1 \pm 0.0$ | $20.1 \pm 1.2$ |
| 35 | $24.2 \pm 1.5$ | $23.8 \pm 2.7$ | $21.5 \pm 0.8$ | $28.1 \pm 0.0$ | $21.2 \pm 1.5$ |
| 40 | $29.6 \pm 1.6$ | $29.7 \pm 2.1$ | $26.1 \pm 1.2$ | $33.1 \pm 0.0$ | $24.5 \pm 2.1$ |
| 45 | $32.8 \pm 1.5$ | $35.9 \pm 2.1$ | $31.6 \pm 1.3$ | $38.1 \pm 0.0$ | $26.8 \pm 1.4$ |
| 50 | $36.6 \pm 1.1$ | $41.3 \pm 2.3$ | $37.3 \pm 2.1$ | $43.1 \pm 0.0$ | $30.1 \pm 2.1$ |
| 55 | $41.2 \pm 1.9$ | $47.0 \pm 1.1$ | $42.3 \pm 1.8$ | $48.1 \pm 0.0$ | $32.9 \pm 2.6$ |

| Actual MI | CCMI | DEMI $\alpha = 0.5$ | DEMI $\alpha = 0.25$ | DEMI $\alpha = 0.75$ |
|---|---|---|---|---|
| 0.1 | $0.1 \pm 0.0$ | $0.1 \pm 0.0$ | $-0.6 \pm 0.0$ | $0.5 \pm 0.0$ |
| 5 | $1.7 \pm 0.1$ | $1.8 \pm 0.1$ | $1.0 \pm 0.1$ | $2.3 \pm 0.2$ |
| 10 | $4.2 \pm 1.1$ | $3.7 \pm 0.3$ | $3.1 \pm 0.3$ | $4.1 \pm 0.2$ |
| 15 | $6.1 \pm 0.6$ | $5.9 \pm 0.5$ | $5.4 \pm 0.6$ | $6.0 \pm 0.3$ |
| 20 | $7.5 \pm 0.8$ | $7.6 \pm 0.8$ | $7.4 \pm 0.6$ | $7.8 \pm 0.7$ |
| 25 | $9.2 \pm 3.0$ | $9.4 \pm 1.0$ | $10.0 \pm 0.5$ | $9.3 \pm 0.8$ |
| 30 | $9.9 \pm 2.8$ | $11.4 \pm 1.3$ | $12.3 \pm 0.9$ | $12.0 \pm 1.0$ |
| 35 | $14.8 \pm 5.2$ | $14.7 \pm 1.7$ | $15.4 \pm 0.8$ | $13.0 \pm 1.7$ |
| 40 | $15.0 \pm 6.5$ | $17.7 \pm 1.6$ | $18.7 \pm 1.5$ | $17.8 \pm 2.0$ |
| 45 | $16.2 \pm 3.0$ | $20.9 \pm 1.7$ | $21.1 \pm 1.4$ | $20.6 \pm 1.9$ |
| 50 | $19.2 \pm 3.2$ | $24.3 \pm 2.1$ | $26.4 \pm 2.2$ | $25.4 \pm 1.8$ |
| 55 | $22.8 \pm 3.4$ | $28.7 \pm 1.6$ | $29.0 \pm 1.3$ | $29.9 \pm 1.4$ |

Table 10: MI estimation between 50-d Gaussian variables trained on 80K data samples. The estimation error $(I(x, y) - \hat{I}(x, y))$ and the standard deviation of the estimates are reported.

| Actual MI | SMILE $\tau = \infty$ | SMILE $\tau = 1$ | SMILE $\tau = 5$ | InfoNCE | GM |
|---|---|---|---|---|---|
| 0.1 | $1.4 \pm 0.3$ | $1.3 \pm 0.2$ | $1.5 \pm 0.3$ | $0.5 \pm 0.1$ | $0.1 \pm 0.0$ |
| 5 | $1.4 \pm 0.1$ | $0.5 \pm 0.4$ | $1.0 \pm 0.1$ | $3.3 \pm 0.2$ | $4.6 \pm 0.2$ |
| 10 | $1.9 \pm 0.3$ | $-0.4 \pm 0.6$ | $-0.6 \pm 0.5$ | $4.4 \pm 0.1$ | $5.5 \pm 0.6$ |
| 15 | $2.6 \pm 0.5$ | $1.4 \pm 0.9$ | $-0.1 \pm 1.3$ | $8.3 \pm 0.0$ | $4.5 \pm 0.3$ |
| 20 | $4.3 \pm 0.9$ | $2.8 \pm 0.8$ | $0.5 \pm 1.6$ | $13.1 \pm 0.0$ | $3.5 \pm 0.5$ |
| 25 | $6.3 \pm 1.2$ | $4.2 \pm 2.7$ | $3.6 \pm 2.3$ | $18.1 \pm 0.0$ | $2.9 \pm 0.6$ |
| 30 | $8.5 \pm 3.6$ | $7.8 \pm 2.5$ | $4.6 \pm 2.7$ | $23.1 \pm 0.0$ | $3.2 \pm 0.4$ |
| 35 | $5.9 \pm 7.9$ | $10.4 \pm 4.9$ | $4.5 \pm 6.6$ | $28.1 \pm 0.0$ | $3.4 \pm 0.5$ |
| 40 | $-0.9 \pm 6.1$ | $4.7 \pm 5.3$ | $4.2 \pm 7.2$ | $33.1 \pm 0.0$ | $3.8 \pm 0.6$ |
| 45 | $5.7 \pm 7.0$ | $16.6 \pm 8.0$ | $10.2 \pm 5.3$ | $38.1 \pm 0.0$ | $4.2 \pm 0.9$ |
| 50 | $14.4 \pm 11.4$ | $21.5 \pm 6.4$ | $18.4 \pm 7.6$ | $43.1 \pm 0.0$ | $5.3 \pm 1.0$ |
| 55 | $20.8 \pm 10.2$ | $32.3 \pm 7.4$ | $23.5 \pm 6.7$ | $48.1 \pm 0.0$ | $6.7 \pm 1.0$ |

| Actual MI | CCMI | DEMI $\alpha = 0.5$ | DEMI $\alpha = 0.25$ | DEMI $\alpha = 0.75$ |
|---|---|---|---|---|
| 0.1 | $0.2 \pm 0.0$ | $0.4 \pm 0.1$ | $-0.2 \pm 0.1$ | $0.6 \pm 0.0$ |
| 5 | $1.1 \pm 0.8$ | $0.3 \pm 0.2$ | $-0.5 \pm 0.2$ | $1.1 \pm 0.3$ |
| 10 | $0.0 \pm 0.5$ | $1.1 \pm 0.4$ | $0.4 \pm 0.4$ | $1.5 \pm 0.5$ |
| 15 | $-0.3 \pm 1.8$ | $1.7 \pm 0.4$ | $0.9 \pm 0.9$ | $2.4 \pm 1.2$ |
| 20 | $-3.8 \pm 2.0$ | $3.9 \pm 1.3$ | $2.8 \pm 1.4$ | $4.1 \pm 2.1$ |
| 25 | $-4.9 \pm 2.8$ | $5.3 \pm 2.6$ | $4.8 \pm 0.9$ | $5.8 \pm 1.7$ |
| 30 | $-4.8 \pm 6.3$ | $5.8 \pm 2.2$ | $7.9 \pm 1.3$ | $9.0 \pm 2.4$ |
| 35 | $-8.9 \pm 8.1$ | $9.6 \pm 2.9$ | $9.4 \pm 2.2$ | $10.1 \pm 2.4$ |
| 40 | $-15.3 \pm 7.8$ | $10.4 \pm 4.1$ | $11.6 \pm 3.5$ | $11.3 \pm 4.3$ |
| 45 | $-19.4 \pm 7.7$ | $10.6 \pm 4.3$ | $11.1 \pm 1.9$ | $13.9 \pm 4.3$ |
| 50 | $-19.6 \pm 6.7$ | $14.9 \pm 4.9$ | $15.0 \pm 2.8$ | $18.1 \pm 3.6$ |
| 55 | $-18.6 \pm 12.5$ | $23.2 \pm 1.8$ | $20.2 \pm 3.2$ | $24.1 \pm 3.8$ |

Table 11: MI estimation between 50-d Gaussian variables with cubic transformation trained on 80K data samples. The estimation error $(I(x, y) - \hat{I}(x, y))$ and the standard deviation of the estimates are reported.

| Actual MI | SMILE $\tau = \infty$ | SMILE $\tau = 1$ | SMILE $\tau = 5$ | InfoNCE | GM |
|---|---|---|---|---|---|
| 0.1 | $1.1 \pm 0.2$ | $1.2 \pm 0.2$ | $1.1 \pm 0.2$ | $0.2 \pm 0.0$ | $0.9 \pm 0.4$ |
| 5 | $3.0 \pm 0.1$ | $2.8 \pm 0.2$ | $3.0 \pm 0.2$ | $1.6 \pm 0.1$ | $5.5 \pm 0.6$ |
| 10 | $6.3 \pm 0.4$ | $4.8 \pm 0.4$ | $5.0 \pm 0.2$ | $4.3 \pm 0.0$ | $10.0 \pm 0.5$ |
| 15 | $9.3 \pm 0.2$ | $8.5 \pm 0.7$ | $6.8 \pm 0.3$ | $8.3 \pm 0.0$ | $13.9 \pm 0.4$ |
| 20 | $12.7 \pm 0.4$ | $12.2 \pm 1.0$ | $9.8 \pm 0.4$ | $13.1 \pm 0.0$ | $18.1 \pm 0.4$ |
| 25 | $16.5 \pm 0.7$ | $16.8 \pm 1.2$ | $13.9 \pm 0.8$ | $18.1 \pm 0.0$ | $22.0 \pm 0.4$ |
| 30 | $20.2 \pm 0.7$ | $22.7 \pm 0.6$ | $18.3 \pm 1.1$ | $23.1 \pm 0.0$ | $25.8 \pm 0.8$ |
| 35 | $24.0 \pm 1.0$ | $27.4 \pm 1.8$ | $23.4 \pm 1.4$ | $28.1 \pm 0.0$ | $29.9 \pm 0.8$ |
| 40 | $28.8 \pm 0.7$ | $33.1 \pm 2.1$ | $29.3 \pm 1.3$ | $33.1 \pm 0.0$ | $33.9 \pm 0.8$ |
| 45 | $32.3 \pm 1.0$ | $38.5 \pm 2.3$ | $34.2 \pm 2.4$ | $38.1 \pm 0.0$ | $37.4 \pm 1.0$ |
| 50 | $37.7 \pm 1.4$ | $44.8 \pm 1.3$ | $41.0 \pm 1.2$ | $43.1 \pm 0.0$ | $41.7 \pm 1.5$ |
| 55 | $41.7 \pm 1.5$ | $50.4 \pm 2.2$ | $46.8 \pm 1.8$ | $48.1 \pm 0.0$ | $45.3 \pm 0.8$ |

| Actual MI | CCMI | DEMI $\alpha = 0.5$ | DEMI $\alpha = 0.25$ | DEMI $\alpha = 0.75$ |
|---|---|---|---|---|
| 0.1 | $0.2 \pm 0.0$ | $0.2 \pm 0.0$ | $-0.5 \pm 0.0$ | $0.5 \pm 0.0$ |
| 5 | $2.2 \pm 0.1$ | $2.0 \pm 0.1$ | $1.3 \pm 0.1$ | $2.5 \pm 0.3$ |
| 10 | $4.5 \pm 0.8$ | $4.1 \pm 0.2$ | $3.5 \pm 0.2$ | $4.4 \pm 0.3$ |
| 15 | $6.0 \pm 0.9$ | $6.0 \pm 0.6$ | $5.7 \pm 0.4$ | $6.3 \pm 0.4$ |
| 20 | $8.2 \pm 1.6$ | $7.7 \pm 0.8$ | $7.4 \pm 0.7$ | $8.5 \pm 0.7$ |
| 25 | $8.8 \pm 2.4$ | $10.1 \pm 1.0$ | $10.5 \pm 1.1$ | $10.6 \pm 1.2$ |
| 30 | $11.9 \pm 2.6$ | $13.0 \pm 1.4$ | $13.4 \pm 0.7$ | $14.2 \pm 1.1$ |
| 35 | $14.7 \pm 2.9$ | $17.6 \pm 1.4$ | $17.0 \pm 1.0$ | $18.3 \pm 1.4$ |
| 40 | $18.4 \pm 3.3$ | $20.9 \pm 2.2$ | $20.7 \pm 1.1$ | $21.2 \pm 1.6$ |
| 45 | $23.0 \pm 3.9$ | $24.0 \pm 2.1$ | $25.4 \pm 1.5$ | $24.7 \pm 2.2$ |
| 50 | $26.1 \pm 3.1$ | $28.5 \pm 1.5$ | $29.8 \pm 0.7$ | $30.2 \pm 1.2$ |
| 55 | $30.4 \pm 2.5$ | $34.1 \pm 1.9$ | $34.1 \pm 0.9$ | $34.2 \pm 2.3$ |

Table 12: MI estimation between 50-d Gaussian variables trained on 32K data samples. The estimation error $(I(x, y) - \hat{I}(x, y))$ and the standard deviation of the estimates are reported.

| Actual MI | SMILE $\tau = \infty$ | SMILE $\tau = 1$ | SMILE $\tau = 5$ | InfoNCE | GM |
|---|---|---|---|---|---|
| 0.1 | $2.2 \pm 0.8$ | $2.2 \pm 0.9$ | $1.9 \pm 0.8$ | $1.5 \pm 0.2$ | $0.1 \pm 0.0$ |
| 5 | $1.7 \pm 0.1$ | $0.6 \pm 0.4$ | $1.2 \pm 0.2$ | $5.7 \pm 0.3$ | $5.0 \pm 0.0$ |
| 10 | $2.4 \pm 0.2$ | $0.3 \pm 0.8$ | $-0.1 \pm 0.5$ | $4.5 \pm 0.1$ | $9.6 \pm 0.1$ |
| 15 | $3.3 \pm 0.7$ | $1.5 \pm 0.8$ | $0.2 \pm 0.8$ | $8.4 \pm 0.1$ | $13.2 \pm 0.4$ |
| 20 | $5.1 \pm 1.2$ | $3.3 \pm 1.5$ | $0.7 \pm 1.0$ | $13.1 \pm 0.0$ | $15.6 \pm 0.7$ |
| 25 | $5.9 \pm 2.6$ | $5.7 \pm 3.8$ | $0.5 \pm 4.2$ | $18.1 \pm 0.0$ | $16.3 \pm 0.8$ |
| 30 | $3.7 \pm 2.3$ | $5.4 \pm 0.6$ | $1.6 \pm 1.5$ | $23.1 \pm 0.0$ | $17.2 \pm 1.8$ |
| 35 | $7.3 \pm 1.7$ | $9.3 \pm 3.6$ | $5.7 \pm 2.6$ | $28.1 \pm 0.0$ | $18.0 \pm 1.1$ |
| 40 | $10.3 \pm 1.7$ | $16.7 \pm 0.7$ | $11.8 \pm 1.8$ | $33.1 \pm 0.0$ | $18.6 \pm 1.3$ |
| 45 | $15.0 \pm 1.5$ | $21.0 \pm 1.5$ | $16.5 \pm 2.9$ | $38.1 \pm 0.0$ | $20.7 \pm 1.6$ |
| 50 | $18.8 \pm 1.2$ | $26.9 \pm 1.0$ | $21.3 \pm 2.3$ | $43.1 \pm 0.0$ | $21.6 \pm 1.2$ |
| 55 | $24.4 \pm 1.1$ | $31.6 \pm 1.1$ | $28.4 \pm 0.8$ | $48.1 \pm 0.0$ | $23.8 \pm 2.1$ |

| Actual MI | CCMI | DEMI $\alpha = 0.5$ | DEMI $\alpha = 0.25$ | DEMI $\alpha = 0.75$ |
|---|---|---|---|---|
| 0.1 | $11.6 \pm 1.4$ | $1.3 \pm 0.2$ | $0.8 \pm 0.1$ | $0.9 \pm 0.1$ |
| 5 | $74.3 \pm 3.9$ | $0.7 \pm 0.3$ | $0.1 \pm 0.1$ | $1.2 \pm 0.2$ |
| 10 | $81.4 \pm 4.2$ | $1.0 \pm 0.4$ | $0.1 \pm 0.7$ | $1.6 \pm 0.6$ |
| 15 | $73.3 \pm 3.3$ | $2.0 \pm 0.8$ | $1.1 \pm 0.9$ | $2.7 \pm 1.1$ |
| 20 | $72.5 \pm 4.3$ | $4.6 \pm 1.0$ | $3.3 \pm 1.5$ | $5.0 \pm 0.8$ |
| 25 | $76.4 \pm 6.2$ | $6.8 \pm 1.1$ | $6.5 \pm 1.4$ | $7.6 \pm 1.9$ |
| 30 | $79.8 \pm 5.9$ | $10.3 \pm 1.6$ | $10.3 \pm 1.9$ | $10.9 \pm 1.5$ |
| 35 | $89.4 \pm 5.5$ | $15.3 \pm 1.7$ | $13.3 \pm 1.6$ | $15.5 \pm 2.1$ |
| 40 | $94.9 \pm 3.7$ | $16.4 \pm 3.9$ | $17.9 \pm 1.9$ | $19.3 \pm 1.8$ |
| 45 | $100.1 \pm 7.9$ | $23.3 \pm 1.9$ | $22.4 \pm 3.1$ | $24.9 \pm 1.4$ |
| 50 | $99.5 \pm 8.5$ | $26.5 \pm 2.2$ | $26.3 \pm 2.1$ | $28.2 \pm 2.2$ |
| 55 | $105.1 \pm 6.7$ | $34.4 \pm 1.7$ | $33.4 \pm 1.2$ | $34.2 \pm 1.7$ |

Table 13: MI estimation between 50-d Gaussian variables with cubic transformation trained on 32K data samples. The estimation error $(I(x, y) - \hat{I}(x, y))$ and the standard deviation of the estimates are reported.

| Actual MI | SMILE $\tau = \infty$ | SMILE $\tau = 1$ | SMILE $\tau = 5$ | InfoNCE | GM |
|---|---|---|---|---|---|
| 0.1 | $2.0 \pm 0.7$ | $2.2 \pm 0.5$ | $2.2 \pm 0.5$ | $0.5 \pm 0.0$ | $1.4 \pm 0.6$ |
| 5 | $2.7 \pm 0.1$ | $3.4 \pm 0.4$ | $2.7 \pm 0.2$ | $1.7 \pm 0.0$ | $4.5 \pm 0.9$ |
| 10 | $5.1 \pm 0.1$ | $6.9 \pm 0.6$ | $4.9 \pm 0.1$ | $4.3 \pm 0.0$ | $9.3 \pm 1.1$ |
| 15 | $8.0 \pm 0.2$ | $10.4 \pm 0.4$ | $7.8 \pm 0.3$ | $8.4 \pm 0.0$ | $13.0 \pm 1.1$ |
| 20 | $11.3 \pm 0.3$ | $15.7 \pm 0.7$ | $11.9 \pm 0.4$ | $13.2 \pm 0.0$ | $17.9 \pm 1.3$ |
| 25 | $15.0 \pm 0.5$ | $21.0 \pm 0.7$ | $17.2 \pm 0.9$ | $18.1 \pm 0.0$ | $23.0 \pm 1.1$ |
| 30 | $18.6 \pm 0.6$ | $26.5 \pm 1.1$ | $22.5 \pm 0.7$ | $23.1 \pm 0.0$ | $26.4 \pm 1.2$ |
| 35 | $22.8 \pm 0.9$ | $32.0 \pm 0.7$ | $27.6 \pm 0.9$ | $28.1 \pm 0.0$ | $31.2 \pm 0.8$ |
| 40 | $26.2 \pm 0.5$ | $37.0 \pm 1.0$ | $32.9 \pm 1.1$ | $33.1 \pm 0.0$ | $36.8 \pm 1.1$ |
| 45 | $31.3 \pm 1.0$ | $42.8 \pm 0.8$ | $38.8 \pm 0.8$ | $38.1 \pm 0.1$ | $40.6 \pm 1.0$ |
| 50 | $35.7 \pm 0.9$ | $47.8 \pm 1.6$ | $44.2 \pm 1.3$ | $43.1 \pm 0.0$ | $46.0 \pm 1.1$ |
| 55 | $41.0 \pm 1.3$ | $53.2 \pm 1.3$ | $50.0 \pm 0.8$ | $48.1 \pm 0.0$ | $51.0 \pm 1.3$ |

| Actual MI | CCMI | DEMI $\alpha = 0.5$ | DEMI $\alpha = 0.25$ | DEMI $\alpha = 0.75$ |
|---|---|---|---|---|
| 0.1 | $0.5 \pm 0.1$ | $0.5 \pm 0.1$ | $-0.1 \pm 0.1$ | $0.6 \pm 0.0$ |
| 5 | $2.7 \pm 0.2$ | $2.4 \pm 0.1$ | $1.7 \pm 0.1$ | $2.6 \pm 0.3$ |
| 10 | $4.9 \pm 0.5$ | $4.0 \pm 0.4$ | $3.7 \pm 0.2$ | $4.1 \pm 0.7$ |
| 15 | $8.1 \pm 1.6$ | $5.9 \pm 0.5$ | $5.8 \pm 0.4$ | $5.9 \pm 0.9$ |
| 20 | $10.4 \pm 1.5$ | $8.6 \pm 0.9$ | $8.6 \pm 0.4$ | $8.3 \pm 0.8$ |
| 25 | $11.6 \pm 1.6$ | $11.9 \pm 1.0$ | $12.1 \pm 0.6$ | $12.0 \pm 0.8$ |
| 30 | $15.6 \pm 1.0$ | $16.6 \pm 0.8$ | $15.5 \pm 0.4$ | $16.2 \pm 0.8$ |
| 35 | $21.4 \pm 6.8$ | $20.3 \pm 0.7$ | $20.1 \pm 0.9$ | $20.8 \pm 1.2$ |
| 40 | $23.3 \pm 2.0$ | $25.7 \pm 0.7$ | $24.8 \pm 1.0$ | $25.3 \pm 0.9$ |
| 45 | $28.3 \pm 2.3$ | $30.3 \pm 1.0$ | $29.3 \pm 0.6$ | $30.1 \pm 0.8$ |
| 50 | $33.3 \pm 3.2$ | $34.6 \pm 1.0$ | $34.4 \pm 0.8$ | $35.1 \pm 1.0$ |
| 55 | $nan \pm nan$ | $38.9 \pm 1.6$ | $38.4 \pm 0.8$ | $39.5 \pm 1.2$ |

Table 14: MI estimation between 100-d Gaussian variables trained on 160K data samples. The estimation error $(I(x, y) - \hat{I}(x, y))$ and the standard deviation of the estimates are reported.

| Actual MI | SMILE $\tau = \infty$ | SMILE $\tau = 1$ | SMILE $\tau = 5$ | InfoNCE | GM |
|---|---|---|---|---|---|
| 0.1 | $1.2 \pm 0.2$ | $1.3 \pm 0.3$ | $1.1 \pm 0.2$ | $0.4 \pm 0.1$ | $0.1 \pm 0.0$ |
| 5 | $1.8 \pm 0.2$ | $0.8 \pm 0.2$ | $1.4 \pm 0.1$ | $2.5 \pm 0.3$ | $5.0 \pm 0.0$ |
| 10 | $2.6 \pm 0.1$ | $1.1 \pm 0.5$ | $0.9 \pm 0.4$ | $4.7 \pm 0.1$ | $9.7 \pm 0.1$ |
| 15 | $3.9 \pm 0.4$ | $2.1 \pm 0.8$ | $0.4 \pm 1.0$ | $8.4 \pm 0.0$ | $13.6 \pm 0.5$ |
| 20 | $5.8 \pm 0.6$ | $3.7 \pm 1.0$ | $1.3 \pm 1.2$ | $13.2 \pm 0.0$ | $15.3 \pm 0.8$ |
| 25 | $9.0 \pm 1.1$ | $8.2 \pm 1.2$ | $3.9 \pm 1.6$ | $18.1 \pm 0.0$ | $15.2 \pm 0.8$ |
| 30 | $11.4 \pm 1.6$ | $12.8 \pm 1.6$ | $8.6 \pm 1.7$ | $23.1 \pm 0.0$ | $15.9 \pm 0.8$ |
| 35 | $15.6 \pm 1.5$ | $14.6 \pm 2.2$ | $12.5 \pm 1.4$ | $28.1 \pm 0.0$ | $16.3 \pm 0.9$ |
| 40 | $15.7 \pm 2.4$ | $17.5 \pm 4.1$ | $15.3 \pm 1.7$ | $33.1 \pm 0.0$ | $16.1 \pm 1.0$ |
| 45 | $20.3 \pm 2.2$ | $23.8 \pm 3.9$ | $18.6 \pm 2.8$ | $38.1 \pm 0.0$ | $16.1 \pm 0.9$ |
| 50 | $23.2 \pm 5.6$ | $21.2 \pm 5.7$ | $20.9 \pm 2.6$ | $43.1 \pm 0.0$ | $16.9 \pm 1.3$ |
| 55 | $26.3 \pm 4.5$ | $29.3 \pm 2.9$ | $25.9 \pm 5.0$ | $48.1 \pm 0.0$ | $17.7 \pm 1.4$ |
| 60 | $29.3 \pm 4.1$ | $28.7 \pm 5.8$ | $29.0 \pm 2.8$ | $53.1 \pm 0.0$ | $17.9 \pm 1.8$ |
| 65 | $34.0 \pm 6.1$ | $35.7 \pm 4.5$ | $33.2 \pm 3.6$ | $58.1 \pm 0.0$ | $18.7 \pm 1.0$ |
| 70 | $37.0 \pm 5.5$ | $40.7 \pm 4.3$ | $30.8 \pm 4.4$ | $63.1 \pm 0.0$ | $20.2 \pm 1.3$ |
| 75 | $35.7 \pm 8.6$ | $38.5 \pm 9.1$ | $38.1 \pm 9.3$ | $68.1 \pm 0.0$ | $21.3 \pm 2.0$ |
| 80 | $34.1 \pm 13.6$ | $44.4 \pm 9.0$ | $40.2 \pm 6.9$ | $73.1 \pm 0.0$ | $21.8 \pm 1.1$ |

| Actual MI | CCMI | DEMI $\alpha = 0.5$ | DEMI $\alpha = 0.25$ | DEMI $\alpha = 0.75$ |
|---|---|---|---|---|
| 0.1 | $0.8 \pm 0.1$ | $0.7 \pm 0.0$ | $0.2 \pm 0.1$ | $0.6 \pm 0.1$ |
| 5 | $1.3 \pm 0.3$ | $1.1 \pm 0.2$ | $0.5 \pm 0.2$ | $1.6 \pm 0.3$ |
| 10 | $1.1 \pm 0.6$ | $1.7 \pm 0.4$ | $1.3 \pm 0.3$ | $2.2 \pm 0.5$ |
| 15 | $1.0 \pm 1.4$ | $2.2 \pm 0.9$ | $2.7 \pm 0.7$ | $3.1 \pm 1.1$ |
| 20 | $2.4 \pm 2.1$ | $4.6 \pm 1.4$ | $4.2 \pm 0.9$ | $5.4 \pm 1.2$ |
| 25 | $3.7 \pm 2.7$ | $6.4 \pm 1.6$ | $8.4 \pm 1.6$ | $7.6 \pm 2.1$ |
| 30 | $1.7 \pm 4.9$ | $6.9 \pm 2.1$ | $11.3 \pm 2.0$ | $10.3 \pm 1.5$ |
| 35 | $0.2 \pm 2.6$ | $7.8 \pm 2.8$ | $15.7 \pm 1.7$ | $11.4 \pm 2.7$ |
| 40 | $2.5 \pm 4.1$ | $9.5 \pm 2.5$ | $16.6 \pm 2.0$ | $14.0 \pm 2.9$ |
| 45 | $2.2 \pm 6.2$ | $10.6 \pm 6.1$ | $21.5 \pm 1.7$ | $17.1 \pm 2.0$ |
| 50 | $2.1 \pm 6.1$ | $10.6 \pm 3.6$ | $22.1 \pm 3.4$ | $17.8 \pm 3.2$ |
| 55 | $8.8 \pm 6.1$ | $16.6 \pm 4.4$ | $26.0 \pm 3.5$ | $21.5 \pm 4.0$ |
| 60 | $3.7 \pm 5.7$ | $16.5 \pm 3.9$ | $28.3 \pm 4.0$ | $23.0 \pm 4.7$ |
| 65 | $12.0 \pm 8.0$ | $20.5 \pm 6.0$ | $30.5 \pm 5.2$ | $27.0 \pm 7.0$ |
| 70 | $10.0 \pm 9.3$ | $22.8 \pm 6.3$ | $34.2 \pm 4.4$ | $28.2 \pm 6.0$ |
| 75 | $12.5 \pm 7.6$ | $24.7 \pm 5.7$ | $39.2 \pm 4.9$ | $32.4 \pm 2.8$ |
| 80 | $12.4 \pm 12.3$ | $28.0 \pm 9.4$ | $41.2 \pm 3.7$ | $39.9 \pm 5.4$ |

Table 15: MI estimation between 100-d Gaussian variables with cubic transformation trained on 160K data samples. The estimation error $(I(x, y) - \hat{I}(x, y))$ and the standard deviation of the estimates are reported.

| Actual MI | SMILE $\tau = \infty$ | SMILE $\tau = 1$ | SMILE $\tau = 5$ | InfoNCE | GM |
|---|---|---|---|---|---|
| 0.1 | $1.1 \pm 0.2$ | $1.2 \pm 0.2$ | $1.1 \pm 0.1$ | $0.1 \pm 0.0$ | $1.2 \pm 0.8$ |
| 5 | $2.9 \pm 0.1$ | $3.9 \pm 0.2$ | $3.0 \pm 0.0$ | $2.1 \pm 0.1$ | $5.7 \pm 1.5$ |
| 10 | $5.4 \pm 0.1$ | $7.7 \pm 0.4$ | $5.5 \pm 0.1$ | $4.4 \pm 0.0$ | $10.3 \pm 1.5$ |
| 15 | $8.3 \pm 0.1$ | $11.2 \pm 0.4$ | $8.5 \pm 0.2$ | $8.4 \pm 0.0$ | $15.6 \pm 1.8$ |
| 20 | $11.5 \pm 0.2$ | $15.6 \pm 0.4$ | $12.2 \pm 0.3$ | $13.2 \pm 0.0$ | $19.8 \pm 1.1$ |
| 25 | $14.6 \pm 0.3$ | $20.4 \pm 1.1$ | $16.8 \pm 0.5$ | $18.1 \pm 0.0$ | $23.8 \pm 0.9$ |
| 30 | $17.9 \pm 0.4$ | $25.7 \pm 0.5$ | $22.0 \pm 0.7$ | $23.1 \pm 0.0$ | $29.3 \pm 1.2$ |
| 35 | $21.9 \pm 0.8$ | $30.9 \pm 0.8$ | $26.8 \pm 0.6$ | $28.1 \pm 0.0$ | $33.2 \pm 0.9$ |
| 40 | $25.9 \pm 0.8$ | $36.3 \pm 0.3$ | $32.1 \pm 0.8$ | $33.1 \pm 0.0$ | $37.8 \pm 1.2$ |
| 45 | $29.7 \pm 1.2$ | $41.7 \pm 0.9$ | $37.4 \pm 0.5$ | $38.1 \pm 0.0$ | $42.0 \pm 1.1$ |
| 50 | $34.7 \pm 1.1$ | $46.8 \pm 0.5$ | $42.5 \pm 0.6$ | $43.1 \pm 0.0$ | $45.8 \pm 1.2$ |
| 55 | $38.7 \pm 1.0$ | $52.0 \pm 0.5$ | $47.7 \pm 1.0$ | $48.1 \pm 0.0$ | $50.5 \pm 1.1$ |
| 60 | $43.5 \pm 1.4$ | $56.8 \pm 0.5$ | $52.9 \pm 0.7$ | $53.1 \pm 0.0$ | $54.4 \pm 1.4$ |
| 65 | $49.5 \pm 1.0$ | $62.1 \pm 0.8$ | $57.7 \pm 0.6$ | $58.1 \pm 0.0$ | $58.2 \pm 0.8$ |
| 70 | $53.4 \pm 1.2$ | $67.0 \pm 1.0$ | $62.9 \pm 0.8$ | $63.1 \pm 0.0$ | $62.8 \pm 1.5$ |
| 75 | $58.6 \pm 1.7$ | $72.2 \pm 1.0$ | $68.1 \pm 0.7$ | $68.1 \pm 0.0$ | $67.0 \pm 1.3$ |
| 80 | $63.4 \pm 1.5$ | $77.2 \pm 1.0$ | $72.8 \pm 0.8$ | $73.1 \pm 0.0$ | $nan \pm nan$ |

| Actual MI | CCMI | DEMI $\alpha = 0.5$ | DEMI $\alpha = 0.25$ | DEMI $\alpha = 0.75$ |
|---|---|---|---|---|
| 0.1 | $0.3 \pm 0.0$ | $0.3 \pm 0.1$ | $-0.3 \pm 0.0$ | $0.5 \pm 0.0$ |
| 5 | $2.5 \pm 0.1$ | $2.4 \pm 0.1$ | $1.8 \pm 0.1$ | $2.9 \pm 0.2$ |
| 10 | $5.0 \pm 0.8$ | $4.7 \pm 0.3$ | $4.2 \pm 0.2$ | $5.2 \pm 0.3$ |
| 15 | $7.3 \pm 0.8$ | $7.2 \pm 0.4$ | $6.3 \pm 0.4$ | $7.3 \pm 0.8$ |
| 20 | $9.1 \pm 1.6$ | $9.0 \pm 0.9$ | $8.2 \pm 0.8$ | $9.7 \pm 0.7$ |
| 25 | $11.7 \pm 1.7$ | $11.7 \pm 0.9$ | $11.2 \pm 0.8$ | $12.9 \pm 1.1$ |
| 30 | $14.1 \pm 2.9$ | $15.2 \pm 0.8$ | $14.2 \pm 0.8$ | $15.7 \pm 0.9$ |
| 35 | $17.3 \pm 2.5$ | $18.3 \pm 1.4$ | $17.7 \pm 1.0$ | $19.4 \pm 1.2$ |
| 40 | $19.0 \pm 4.1$ | $22.9 \pm 0.7$ | $21.8 \pm 0.8$ | $23.5 \pm 1.4$ |
| 45 | $21.4 \pm 1.9$ | $27.0 \pm 1.4$ | $25.9 \pm 1.1$ | $27.0 \pm 1.3$ |
| 50 | $25.9 \pm 1.6$ | $31.4 \pm 1.0$ | $31.0 \pm 0.8$ | $31.2 \pm 1.1$ |
| 55 | $32.1 \pm 4.8$ | $35.1 \pm 1.9$ | $34.8 \pm 1.2$ | $35.2 \pm 1.9$ |
| 60 | $35.5 \pm 3.7$ | $39.9 \pm 2.6$ | $39.7 \pm 1.3$ | $39.4 \pm 1.7$ |
| 65 | $38.0 \pm 2.3$ | $44.4 \pm 1.6$ | $44.0 \pm 2.2$ | $43.4 \pm 2.0$ |
| 70 | $44.0 \pm 3.0$ | $49.4 \pm 1.3$ | $48.1 \pm 1.3$ | $48.1 \pm 1.8$ |
| 75 | $46.1 \pm 1.8$ | $53.2 \pm 1.0$ | $52.7 \pm 1.2$ | $53.5 \pm 1.9$ |
| 80 | $52.6 \pm 1.7$ | $58.8 \pm 1.5$ | $57.4 \pm 2.9$ | $58.5 \pm 1.5$ |

Table 16: MI estimation between 100-d Gaussian variables trained on 80K data samples. The estimation error $(I(x,y) - \hat{I}(x,y))$ and the standard deviation of the estimates are reported.

| Actual MI | SMILE $\tau = \infty$ | SMILE $\tau = 1$ | SMILE $\tau = 5$ | InfoNCE | GM |
|---|---|---|---|---|---|
| 0.1 | $2.1 \pm 0.6$ | $2.3 \pm 0.7$ | $2.3 \pm 0.3$ | $0.9 \pm 0.1$ | $0.1 \pm 0.0$ |
| 5 | $2.0 \pm 0.1$ | $1.0 \pm 0.4$ | $1.7 \pm 0.2$ | $4.5 \pm 0.2$ | $5.0 \pm 0.0$ |
| 10 | $3.1 \pm 0.2$ | $1.4 \pm 0.5$ | $1.0 \pm 0.4$ | $4.8 \pm 0.1$ | $10.0 \pm 0.0$ |
| 15 | $4.6 \pm 0.2$ | $3.2 \pm 1.0$ | $2.4 \pm 0.6$ | $8.6 \pm 0.0$ | $14.7 \pm 0.1$ |
| 20 | $6.8 \pm 1.1$ | $5.8 \pm 1.3$ | $4.9 \pm 1.2$ | $13.3 \pm 0.0$ | $19.0 \pm 0.2$ |
| 25 | $10.0 \pm 1.2$ | $7.9 \pm 2.0$ | $6.9 \pm 1.7$ | $18.2 \pm 0.0$ | $22.8 \pm 0.7$ |
| 30 | $13.5 \pm 1.8$ | $12.3 \pm 1.3$ | $9.1 \pm 1.4$ | $23.1 \pm 0.0$ | $24.6 \pm 1.0$ |
| 35 | $16.8 \pm 2.3$ | $16.8 \pm 2.1$ | $13.5 \pm 1.5$ | $28.1 \pm 0.0$ | $26.2 \pm 1.4$ |
| 40 | $18.9 \pm 2.7$ | $21.4 \pm 2.3$ | $18.3 \pm 2.2$ | $33.1 \pm 0.0$ | $27.5 \pm 1.1$ |
| 45 | $25.4 \pm 2.6$ | $23.9 \pm 2.4$ | $21.9 \pm 1.9$ | $38.1 \pm 0.0$ | $28.2 \pm 0.9$ |
| 50 | $26.8 \pm 5.9$ | $29.0 \pm 6.0$ | $25.3 \pm 1.7$ | $43.1 \pm 0.0$ | $29.9 \pm 1.1$ |
| 55 | $29.6 \pm 6.7$ | $28.4 \pm 7.0$ | $25.0 \pm 5.1$ | $48.1 \pm 0.0$ | $31.1 \pm 1.5$ |
| 60 | $26.9 \pm 6.4$ | $28.0 \pm 6.6$ | $29.9 \pm 6.9$ | $53.1 \pm 0.0$ | $31.7 \pm 1.3$ |
| 65 | $24.3 \pm 6.1$ | $34.1 \pm 10.6$ | $30.8 \pm 7.4$ | $58.1 \pm 0.0$ | $33.5 \pm 1.6$ |
| 70 | $30.5 \pm 8.5$ | $36.6 \pm 2.1$ | $35.0 \pm 6.7$ | $63.1 \pm 0.0$ | $35.3 \pm 1.2$ |
| 75 | $40.0 \pm 13.0$ | $40.0 \pm 3.1$ | $39.1 \pm 6.7$ | $68.1 \pm 0.0$ | $36.3 \pm 0.9$ |
| 80 | $41.1 \pm 9.2$ | $48.8 \pm 8.8$ | $44.2 \pm 5.6$ | $73.1 \pm 0.0$ | $38.2 \pm 1.4$ |

| Actual MI | CCMI | DEMI $\alpha = 0.5$ | DEMI $\alpha = 0.25$ | DEMI $\alpha = 0.75$ |
|---|---|---|---|---|
| 0.1 | $0.8 \pm 0.1$ | $1.9 \pm 0.2$ | $1.2 \pm 0.1$ | $1.0 \pm 0.1$ |
| 5 | $1.3 \pm 0.3$ | $1.6 \pm 0.2$ | $1.0 \pm 0.2$ | $1.8 \pm 0.3$ |
| 10 | $1.1 \pm 0.6$ | $2.1 \pm 0.6$ | $1.4 \pm 0.3$ | $2.9 \pm 0.4$ |
| 15 | $1.0 \pm 1.4$ | $3.5 \pm 0.9$ | $2.4 \pm 0.4$ | $4.7 \pm 0.8$ |
| 20 | $2.4 \pm 2.1$ | $5.4 \pm 1.0$ | $4.5 \pm 1.2$ | $7.6 \pm 1.5$ |
| 25 | $3.7 \pm 2.7$ | $8.9 \pm 1.2$ | $8.4 \pm 0.9$ | $10.2 \pm 1.2$ |
| 30 | $1.7 \pm 4.9$ | $9.8 \pm 1.8$ | $10.8 \pm 1.2$ | $13.3 \pm 2.1$ |
| 35 | $0.2 \pm 2.6$ | $13.6 \pm 1.8$ | $14.8 \pm 1.9$ | $18.0 \pm 1.9$ |
| 40 | $2.5 \pm 4.1$ | $16.0 \pm 1.9$ | $18.1 \pm 1.4$ | $20.0 \pm 2.2$ |
| 45 | $2.2 \pm 6.2$ | $17.8 \pm 3.5$ | $22.0 \pm 1.4$ | $23.9 \pm 1.7$ |
| 50 | $2.1 \pm 6.1$ | $22.2 \pm 2.8$ | $26.0 \pm 2.4$ | $26.6 \pm 2.4$ |
| 55 | $8.8 \pm 6.1$ | $23.3 \pm 3.2$ | $30.1 \pm 2.7$ | $30.8 \pm 3.3$ |
| 60 | $3.7 \pm 5.7$ | $26.0 \pm 5.5$ | $33.2 \pm 2.5$ | $33.4 \pm 3.4$ |
| 65 | $12.0 \pm 8.0$ | $33.5 \pm 2.8$ | $36.2 \pm 4.0$ | $39.0 \pm 2.5$ |
| 70 | $10.0 \pm 9.3$ | $35.1 \pm 3.1$ | $42.1 \pm 2.4$ | $39.9 \pm 2.3$ |
| 75 | $12.5 \pm 7.6$ | $37.8 \pm 3.4$ | $45.5 \pm 2.1$ | $45.7 \pm 3.5$ |
| 80 | $12.4 \pm 12.3$ | $44.1 \pm 4.2$ | $48.7 \pm 3.5$ | $51.0 \pm 3.4$ |

Table 17: MI estimation between 100-d Gaussian variables with cubic transformation trained on 80K data samples. The estimation error $(I(x, y) - \hat{I}(x, y))$ and the standard deviation of the estimates are reported.

| Actual MI | SMILE $\tau = \infty$ | SMILE $\tau = 1$ | SMILE $\tau = 5$ | InfoNCE | GM |
|---|---|---|---|---|---|
| 0.1 | $2.3 \pm 0.4$ | $2.4 \pm 0.4$ | $2.1 \pm 0.5$ | $0.3 \pm 0.0$ | $5.5 \pm 2.0$ |
| 5 | $3.5 \pm 0.1$ | $4.5 \pm 0.3$ | $3.4 \pm 0.1$ | $2.2 \pm 0.1$ | $9.5 \pm 1.9$ |
| 10 | $5.8 \pm 0.1$ | $8.2 \pm 0.3$ | $5.9 \pm 0.2$ | $4.5 \pm 0.0$ | $13.6 \pm 2.2$ |
| 15 | $8.6 \pm 0.2$ | $12.9 \pm 0.5$ | $9.4 \pm 0.3$ | $8.5 \pm 0.0$ | $18.6 \pm 1.6$ |
| 20 | $11.7 \pm 0.1$ | $17.3 \pm 0.5$ | $13.6 \pm 0.3$ | $13.2 \pm 0.0$ | $22.8 \pm 1.8$ |
| 25 | $15.3 \pm 0.3$ | $22.3 \pm 0.4$ | $18.2 \pm 0.3$ | $18.1 \pm 0.0$ | $27.9 \pm 1.7$ |
| 30 | $19.1 \pm 0.4$ | $27.1 \pm 0.9$ | $23.3 \pm 0.4$ | $23.1 \pm 0.0$ | $32.2 \pm 1.5$ |
| 35 | $23.4 \pm 0.5$ | $32.4 \pm 0.5$ | $28.2 \pm 0.8$ | $28.1 \pm 0.0$ | $36.5 \pm 0.9$ |
| 40 | $27.6 \pm 0.4$ | $37.7 \pm 0.4$ | $33.9 \pm 0.7$ | $33.1 \pm 0.0$ | $41.3 \pm 1.2$ |
| 45 | $32.4 \pm 0.8$ | $42.8 \pm 0.8$ | $38.8 \pm 0.5$ | $38.1 \pm 0.0$ | $44.7 \pm 2.1$ |
| 50 | $37.0 \pm 0.7$ | $47.5 \pm 0.7$ | $43.9 \pm 0.7$ | $43.1 \pm 0.0$ | $50.0 \pm 1.4$ |
| 55 | $42.4 \pm 1.1$ | $53.2 \pm 0.8$ | $49.2 \pm 0.8$ | $48.1 \pm 0.0$ | $55.1 \pm 1.1$ |
| 60 | $46.6 \pm 1.0$ | $58.5 \pm 0.7$ | $54.2 \pm 0.8$ | $53.1 \pm 0.0$ | $59.8 \pm 1.5$ |
| 65 | $51.6 \pm 1.0$ | $63.4 \pm 0.8$ | $59.3 \pm 0.8$ | $58.1 \pm 0.0$ | $65.0 \pm 0.9$ |
| 70 | $56.5 \pm 0.7$ | $69.3 \pm 0.7$ | $64.4 \pm 0.9$ | $63.1 \pm 0.1$ | $68.7 \pm 1.4$ |
| 75 | $61.7 \pm 1.0$ | $73.3 \pm 0.6$ | $69.8 \pm 0.4$ | $68.1 \pm 0.0$ | $72.9 \pm 1.3$ |
| 80 | $67.3 \pm 0.6$ | $79.3 \pm 0.6$ | $74.9 \pm 0.5$ | $73.1 \pm 0.1$ | $78.4 \pm 1.3$ |

| Actual MI | CCMI | DEMI $\alpha = 0.5$ | DEMI $\alpha = 0.25$ | DEMI $\alpha = 0.75$ |
|---|---|---|---|---|
| 0.1 | $0.8 \pm 0.1$ | $0.8 \pm 0.1$ | $0.2 \pm 0.1$ | $0.6 \pm 0.0$ |
| 5 | $3.1 \pm 0.2$ | $2.9 \pm 0.1$ | $2.3 \pm 0.2$ | $2.9 \pm 0.4$ |
| 10 | $5.1 \pm 0.4$ | $4.8 \pm 0.2$ | $4.3 \pm 0.2$ | $4.8 \pm 1.1$ |
| 15 | $7.3 \pm 1.2$ | $6.6 \pm 0.9$ | $6.3 \pm 0.4$ | $7.0 \pm 1.1$ |
| 20 | $10.4 \pm 1.5$ | $9.6 \pm 1.1$ | $8.7 \pm 0.6$ | $9.9 \pm 0.9$ |
| 25 | $11.1 \pm 1.8$ | $12.7 \pm 0.6$ | $11.9 \pm 1.0$ | $13.7 \pm 0.6$ |
| 30 | $14.7 \pm 2.1$ | $16.2 \pm 1.1$ | $15.4 \pm 1.3$ | $16.1 \pm 1.0$ |
| 35 | $18.4 \pm 2.7$ | $20.4 \pm 1.0$ | $19.4 \pm 0.7$ | $20.6 \pm 1.4$ |
| 40 | $22.7 \pm 3.9$ | $24.7 \pm 0.6$ | $23.2 \pm 0.8$ | $24.6 \pm 1.4$ |
| 45 | $25.6 \pm 2.0$ | $28.6 \pm 1.0$ | $28.3 \pm 1.1$ | $29.6 \pm 1.1$ |
| 50 | $29.8 \pm 3.6$ | $32.8 \pm 1.8$ | $32.1 \pm 0.8$ | $34.0 \pm 1.3$ |
| 55 | $34.2 \pm 3.5$ | $37.5 \pm 1.4$ | $36.9 \pm 1.3$ | $38.3 \pm 1.5$ |
| 60 | $39.8 \pm 3.0$ | $42.1 \pm 0.9$ | $42.0 \pm 1.6$ | $42.6 \pm 1.3$ |
| 65 | $43.4 \pm 4.0$ | $47.2 \pm 1.6$ | $47.3 \pm 1.0$ | $47.8 \pm 1.8$ |
| 70 | $46.6 \pm 2.3$ | $52.1 \pm 1.3$ | $51.6 \pm 0.9$ | $52.5 \pm 1.3$ |
| 75 | $51.5 \pm 3.4$ | $57.1 \pm 1.3$ | $56.3 \pm 1.0$ | $57.9 \pm 1.8$ |
| 80 | $56.4 \pm 2.5$ | $62.1 \pm 0.9$ | $61.4 \pm 0.8$ | $62.4 \pm 1.5$ |

Table 18: MI estimation between 100-d Gaussian variables trained on 32K data samples. The estimation error $(I(x,y) - \hat{I}(x,y))$ and the standard deviation of the estimates are reported.

| Actual MI | SMILE $\tau = \infty$ | SMILE $\tau = 1$ | SMILE $\tau = 5$ | InfoNCE | GM |
|---|---|---|---|---|---|
| 0.1 | $3.6 \pm 1.2$ | $3.5 \pm 1.2$ | $3.7 \pm 0.6$ | $2.7 \pm 0.2$ | $0.1 \pm 0.0$ |
| 5 | $3.0 \pm 0.2$ | $1.2 \pm 0.2$ | $2.3 \pm 0.2$ | $6.2 \pm 0.2$ | $5.0 \pm 0.0$ |
| 10 | $4.2 \pm 0.3$ | $2.1 \pm 0.6$ | $2.1 \pm 0.5$ | $5.2 \pm 0.1$ | $10.0 \pm 0.0$ |
| 15 | $6.4 \pm 0.4$ | $4.0 \pm 0.9$ | $2.9 \pm 0.9$ | $8.8 \pm 0.1$ | $15.0 \pm 0.0$ |
| 20 | $9.0 \pm 0.5$ | $7.0 \pm 1.8$ | $5.7 \pm 1.1$ | $13.4 \pm 0.1$ | $20.0 \pm 0.0$ |
| 25 | $11.9 \pm 1.2$ | $10.4 \pm 1.2$ | $7.8 \pm 1.6$ | $18.2 \pm 0.0$ | $25.0 \pm 0.0$ |
| 30 | $15.0 \pm 1.6$ | $14.2 \pm 1.5$ | $10.6 \pm 1.6$ | $23.1 \pm 0.0$ | $29.8 \pm 0.1$ |
| 35 | $19.2 \pm 2.0$ | $16.1 \pm 2.3$ | $15.7 \pm 2.3$ | $28.1 \pm 0.0$ | $34.4 \pm 0.3$ |
| 40 | $17.9 \pm 3.4$ | $18.4 \pm 3.4$ | $14.9 \pm 3.9$ | $33.1 \pm 0.0$ | $38.9 \pm 0.5$ |
| 45 | $18.6 \pm 4.2$ | $22.0 \pm 2.7$ | $17.7 \pm 1.8$ | $38.1 \pm 0.0$ | $43.4 \pm 0.3$ |
| 50 | $22.9 \pm 1.8$ | $24.9 \pm 3.6$ | $23.1 \pm 1.1$ | $43.1 \pm 0.0$ | $47.4 \pm 0.6$ |
| 55 | $25.9 \pm 2.2$ | $30.2 \pm 3.8$ | $27.2 \pm 2.9$ | $48.1 \pm 0.0$ | $51.3 \pm 0.7$ |
| 60 | $30.1 \pm 2.1$ | $36.2 \pm 2.9$ | $31.8 \pm 3.2$ | $53.1 \pm 0.0$ | $54.9 \pm 1.5$ |
| 65 | $35.4 \pm 1.0$ | $40.4 \pm 3.2$ | $36.3 \pm 2.8$ | $58.1 \pm 0.0$ | $58.2 \pm 1.7$ |
| 70 | $38.9 \pm 2.5$ | $47.0 \pm 1.7$ | $43.1 \pm 1.6$ | $63.1 \pm 0.0$ | $61.7 \pm 1.4$ |
| 75 | $44.6 \pm 0.8$ | $52.7 \pm 0.4$ | $48.5 \pm 0.9$ | $68.1 \pm 0.0$ | $65.2 \pm 2.2$ |
| 80 | $49.1 \pm 1.8$ | $57.6 \pm 1.2$ | $54.1 \pm 0.6$ | $73.1 \pm 0.0$ | $68.9 \pm 1.8$ |

| Actual MI | CCMI | DEMI $\alpha = 0.5$ | DEMI $\alpha = 0.25$ | DEMI $\alpha = 0.75$ |
|---|---|---|---|---|
| 0.1 | $19.6 \pm 1.6$ | $4.3 \pm 0.1$ | $3.7 \pm 0.3$ | $1.6 \pm 0.1$ |
| 5 | $63.3 \pm 3.7$ | $1.8 \pm 0.3$ | $1.4 \pm 0.2$ | $2.0 \pm 0.3$ |
| 10 | $54.3 \pm 3.6$ | $2.1 \pm 0.5$ | $1.5 \pm 0.4$ | $1.9 \pm 0.9$ |
| 15 | $51.8 \pm 5.3$ | $3.8 \pm 0.9$ | $3.7 \pm 0.7$ | $4.5 \pm 0.9$ |
| 20 | $52.0 \pm 3.4$ | $6.9 \pm 1.0$ | $6.1 \pm 0.9$ | $7.0 \pm 1.2$ |
| 25 | $60.7 \pm 3.6$ | $10.1 \pm 0.9$ | $9.3 \pm 0.8$ | $11.2 \pm 1.0$ |
| 30 | $64.6 \pm 3.5$ | $12.9 \pm 1.5$ | $12.6 \pm 0.7$ | $15.1 \pm 1.4$ |
| 35 | $69.2 \pm 4.4$ | $17.6 \pm 1.1$ | $16.7 \pm 1.1$ | $19.1 \pm 1.1$ |
| 40 | $74.0 \pm 4.8$ | $21.4 \pm 1.8$ | $20.7 \pm 1.0$ | $23.0 \pm 1.3$ |
| 45 | $78.7 \pm 2.8$ | $25.2 \pm 1.5$ | $24.5 \pm 1.1$ | $27.4 \pm 1.1$ |
| 50 | $82.0 \pm 4.3$ | $30.8 \pm 1.2$ | $29.4 \pm 1.1$ | $31.5 \pm 1.7$ |
| 55 | $87.6 \pm 6.9$ | $34.5 \pm 1.9$ | $33.8 \pm 1.3$ | $38.0 \pm 1.2$ |
| 60 | $92.7 \pm 4.6$ | $39.0 \pm 2.2$ | $39.2 \pm 1.2$ | $40.8 \pm 1.4$ |
| 65 | $96.4 \pm 6.0$ | $42.8 \pm 2.5$ | $43.7 \pm 1.5$ | $46.4 \pm 1.6$ |
| 70 | $100.6 \pm 5.4$ | $48.5 \pm 1.6$ | $47.7 \pm 1.3$ | $51.6 \pm 1.3$ |
| 75 | $103.9 \pm 5.2$ | $52.6 \pm 2.0$ | $53.9 \pm 2.0$ | $54.8 \pm 1.4$ |
| 80 | $109.3 \pm 6.0$ | $58.7 \pm 0.8$ | $59.2 \pm 1.3$ | $61.8 \pm 1.2$ |

Table 19: MI estimation between 100-d Gaussian variables with cubic transformation trained on 32K data samples. The estimation error $(I(x, y) - \hat{I}(x, y))$ and the standard deviation of the estimates are reported.

| Actual MI | SMILE $\tau = \infty$ | SMILE $\tau = 1$ | SMILE $\tau = 5$ | InfoNCE | GM |
|---|---|---|---|---|---|
| 0.1 | $2.7 \pm 0.6$ | $2.7 \pm 1.0$ | $2.8 \pm 0.5$ | $0.8 \pm 0.1$ | $2.9 \pm 2.7$ |
| 5 | $4.2 \pm 0.2$ | $5.0 \pm 0.5$ | $4.2 \pm 0.3$ | $2.4 \pm 0.0$ | $6.1 \pm 0.9$ |
| 10 | $6.6 \pm 0.2$ | $9.3 \pm 0.5$ | $6.8 \pm 0.2$ | $4.6 \pm 0.0$ | $9.8 \pm 1.6$ |
| 15 | $9.7 \pm 0.2$ | $13.7 \pm 0.6$ | $10.5 \pm 0.4$ | $8.6 \pm 0.1$ | $13.4 \pm 1.6$ |
| 20 | $13.2 \pm 0.2$ | $18.6 \pm 0.4$ | $14.9 \pm 0.4$ | $13.4 \pm 0.1$ | $17.3 \pm 1.7$ |
| 25 | $17.0 \pm 0.3$ | $23.2 \pm 0.6$ | $19.5 \pm 0.5$ | $18.2 \pm 0.1$ | $21.7 \pm 1.2$ |
| 30 | $21.1 \pm 0.2$ | $28.5 \pm 0.7$ | $24.4 \pm 0.4$ | $23.2 \pm 0.1$ | $26.1 \pm 2.5$ |
| 35 | $25.5 \pm 0.4$ | $33.5 \pm 0.5$ | $29.5 \pm 0.5$ | $28.2 \pm 0.1$ | $30.7 \pm 2.1$ |
| 40 | $30.0 \pm 0.5$ | $38.6 \pm 0.4$ | $34.3 \pm 0.5$ | $33.1 \pm 0.0$ | $36.4 \pm 1.9$ |
| 45 | $34.7 \pm 0.4$ | $43.5 \pm 0.7$ | $39.3 \pm 0.7$ | $38.1 \pm 0.0$ | $40.4 \pm 1.9$ |
| 50 | $39.4 \pm 0.4$ | $48.6 \pm 0.5$ | $44.6 \pm 0.4$ | $43.1 \pm 0.1$ | $46.0 \pm 1.2$ |
| 55 | $44.0 \pm 0.4$ | $53.5 \pm 0.8$ | $50.1 \pm 0.6$ | $48.1 \pm 0.1$ | $49.1 \pm 1.7$ |
| 60 | $49.0 \pm 1.0$ | $59.1 \pm 0.7$ | $54.5 \pm 0.7$ | $53.2 \pm 0.1$ | $54.6 \pm 1.2$ |
| 65 | $53.5 \pm 0.5$ | $63.8 \pm 0.8$ | $60.0 \pm 0.6$ | $58.2 \pm 0.1$ | $58.4 \pm 1.5$ |
| 70 | $58.5 \pm 0.7$ | $69.2 \pm 0.5$ | $65.1 \pm 0.8$ | $63.1 \pm 0.0$ | $64.2 \pm 0.9$ |
| 75 | $63.4 \pm 0.9$ | $74.3 \pm 0.8$ | $70.0 \pm 0.5$ | $68.1 \pm 0.1$ | $67.7 \pm 2.0$ |
| 80 | $68.9 \pm 0.8$ | $79.4 \pm 1.3$ | $75.7 \pm 0.7$ | $73.1 \pm 0.1$ | $74.1 \pm 1.6$ |

| Actual MI | CCMI | DEMI $\alpha = 0.5$ | DEMI $\alpha = 0.25$ | DEMI $\alpha = 0.75$ |
|---|---|---|---|---|
| 0.1 | $2.0 \pm 0.3$ | $2.0 \pm 0.3$ | $1.6 \pm 0.1$ | $0.8 \pm 0.1$ |
| 5 | $3.8 \pm 0.1$ | $3.1 \pm 0.3$ | $2.8 \pm 0.3$ | $2.7 \pm 0.4$ |
| 10 | $6.6 \pm 0.7$ | $4.8 \pm 0.6$ | $4.9 \pm 0.2$ | $4.0 \pm 0.5$ |
| 15 | $8.7 \pm 2.4$ | $7.0 \pm 0.6$ | $7.3 \pm 0.5$ | $6.6 \pm 0.6$ |
| 20 | $10.5 \pm 2.3$ | $10.5 \pm 0.6$ | $10.3 \pm 0.6$ | $10.3 \pm 0.7$ |
| 25 | $14.2 \pm 2.2$ | $14.5 \pm 0.9$ | $14.2 \pm 0.5$ | $13.2 \pm 1.0$ |
| 30 | $16.3 \pm 2.2$ | $17.9 \pm 0.8$ | $17.6 \pm 0.7$ | $17.3 \pm 1.2$ |
| 35 | $20.7 \pm 2.1$ | $22.7 \pm 0.7$ | $22.1 \pm 1.1$ | $22.5 \pm 0.7$ |
| 40 | $24.7 \pm 2.4$ | $27.6 \pm 1.1$ | $26.7 \pm 0.9$ | $26.6 \pm 0.7$ |
| 45 | $28.9 \pm 1.9$ | $31.8 \pm 1.1$ | $31.5 \pm 0.6$ | $31.2 \pm 0.9$ |
| 50 | $33.2 \pm 2.7$ | $36.4 \pm 1.4$ | $36.2 \pm 0.9$ | $34.7 \pm 1.2$ |
| 55 | $37.9 \pm 1.5$ | $41.4 \pm 0.8$ | $40.8 \pm 1.1$ | $40.5 \pm 1.2$ |
| 60 | $42.5 \pm 1.3$ | $46.2 \pm 1.5$ | $45.6 \pm 0.9$ | $44.5 \pm 1.6$ |
| 65 | $46.1 \pm 2.9$ | $51.1 \pm 0.7$ | $50.8 \pm 1.1$ | $50.0 \pm 1.0$ |
| 70 | $53.2 \pm 2.6$ | $56.7 \pm 1.4$ | $55.9 \pm 0.9$ | $55.5 \pm 1.1$ |
| 75 | $56.3 \pm 1.7$ | $60.8 \pm 1.2$ | $60.3 \pm 1.0$ | $60.1 \pm 1.0$ |
| 80 | $61.9 \pm 3.2$ | $66.7 \pm 1.5$ | $64.9 \pm 1.0$ | $65.0 \pm 0.8$ |

