# OpenReview forum: "DEMI: Discriminative Estimator of Mutual Information "
_ICLR.cc/2021/Conference — Reject_

### Official Review · AnonReviewer2 · 2020-10-21
**Discriminative Estimator of Mutual Information**

**Rating:** 5
**Confidence:** 5

**Review:**

This paper proposed a discriminative estimator for mutual information, to alleviate the shortcomings of the existing estimators such as MINE and SMILE. A classifier was built to decide whether the sample is drawn from the joint distribution or the independent one (product of marginals). Theoretical justification and experimental results were provided to support the proposed estimator. The paper was written with clarity and easy to follow.

Here are some detailed comments on the technical contribution of this paper:
1) There is a closely related piece of work in the literature (see below). They also proposed a discriminative estimator for KL divergence, with mutual information as a special case. It would be nice if the authors could relate to this existing work, and provide experimental comparison to their estimator.
Mukherjee, Sudipto, Himanshu Asnani, and Sreeram Kannan. "CCMI: Classifier based conditional mutual information estimation." In Uncertainty in Artificial Intelligence, pp. 1083-1093. PMLR, 2020. (https://arxiv.org/pdf/1906.01824.pdf)

2) From Figure 1 right column, we see that all estimators (including the one proposed) underestimate the mutual information when it is high. Could the authors give more analysis and explanation on this phenomenon?

3) It would be nice if the authors could provide experimental results on more realistic datasets, and show the advantage of the proposed estimator when it is used for other downstream tasks. Often, estimating the mutual information is not the end goal, but an intermediate step to achieve other goals (see the MINE paper for examples).

4) A minor point: In equation (10) the last part, it should be (1-z) * log( 1 - q(...) ) instead of (1-z) * ( 1 - log(q(...)) ).

---

> ### Author Response · Authors · 2020-11-25
> **Updated the discussion of related work and expanded experimental evaluation. Thank you for the feedback!**
>
> *1. There is a closely related piece of work in the literature (see below). They also proposed a discriminative estimator for KL divergence, with mutual information as a special case. It would be nice if the authors could relate to this existing work, and provide experimental comparison to their estimator. Mukherjee, Sudipto, Himanshu Asnani, and Sreeram Kannan. "CCMI: Classifier based conditional mutual information estimation." In Uncertainty in Artificial Intelligence, pp. 1083-1093. PMLR, 2020. (https://arxiv.org/pdf/1906.01824.pdf)*
>
> >We thank the reviewer for bringing CCMI to our attention. We expanded our discussion of related methods and empirical evaluation to include CCMI. CCMI improves upon MINE by directly constructing the classifier function to be optimized over the training set (rather than optimizing over arbitrary functions), but it still employs the variational lower bound as the estimate of MI. In particular, the estimate relies both on paired and unpaired data. The term that depends on the unpaired data is asymptotically zero, but the limited data setup leads to increased errors relative to our method (DEMI), as our experiments demonstrate.
>
> *2. From Figure 1 right column, we see that all estimators (including the one proposed) underestimate the mutual information when it is high. Could the authors give more analysis and explanation on this phenomenon?*
>
> >We thank the reviewer for the comment. As we discuss in the Introduction, the estimates that rely on variational lower bounds are bounded from above by O(log N), where N is the number of sample pairs  (Song & Ermon, 2019; McAllester & Stratos, 2020). For the same number of training sample pairs, the estimators become increasingly inaccurate for high MI setups. The accuracy of our estimator is governed by the ability of the classifier to match true log odds values in the test set, which in turn depends on the number of training sample pairs and the capacity of the neural architecture used in training. Thus the quality of our estimates is subject to the classical issues of model capacity and overfitting in deep learning. We leave theoretical analysis (which is closely related to the classifier’s convergence to the true separating boundary for a general hypothesis class) for future work. We also note that for low MI values relative to the size of the training set, our (and other) estimators could overestimate MI, as can be seen in the case of 20-dimensional Gaussian variables in Fig. 1.
>
> *3. It would be nice if the authors could provide experimental results on more realistic datasets, and show the advantage of the proposed estimator when it is used for other downstream tasks. Often, estimating the mutual information is not the end goal, but an intermediate step to achieve other goals (see the MINE paper for examples).*
>
> >Following the reviewers’ feedback, we performed a more realistic experiment on CIFAR10 and CIFAR100. We use different MI estimators including DEMI to perform representation learning based on Deep InfoMax (Hjelm et al., 2018) that relies on mutual information maximization. We have found that DEMI outperforms the other MI estimators on the downstream classification task. We describe this experiment in the updated manuscript.
>
> *4. A minor point: In equation (10)...*
>
> >Thank the reviewer for the catch. We carefully proofread the paper, fixed typos, and improved the figures throughout the paper.

---

### Official Review · AnonReviewer3 · 2020-10-31
**An interesting approach with promising results**

**Rating:** 6
**Confidence:** 3

**Review:**

## Summary:

This paper proposes DEMI, a discriminative approach to estimate mutual information (MI). The main idea is that, instead of learning (generative) distributions of joint and marginals, learning a single likelihood ratio that is discriminative and hence more tractable: a posterior $p(z | x, y)$ trying to distinguish between the joint distribution $p(x, y)$ and the product distribution $p(x)p(y)$. Once the posterior is learned, it can be used to estimate the MI.

## Strength:

- This paper studies a very important problem for the representation learning community -- mutual information has been a very powerful, principled technique for deep representation learning and many applications, but there are many challenges in scalable and accurate, low-variance estimation. Therefore developing an accurate MI estimator is of high importance and significance.
- I find the idea of “lifting” the distribution and converting the MI estimation problem into a discriminative setting interesting, and looks novel. The method makes sense, and the training procedure is very simple, achieving better estimation than the baselines.
- This paper is well-placed and contains comprehensive discussion of recent works about the limitations and research challenges on mutual information estimation. The mathematical connection to existing methods (MINE, InfoNCE, SMILE, etc.) provide an interesting insight.

## Weakness:

- The method only discusses estimation of mutual information, not maximization of MI for representation learning.
- The biggest weakness of this paper would be: experiments. The training data used in the experiments is either low-dimensional or synthetic, so there remains a question about how well this method will scale to a high-dimensional, and challenging deep learning setting. As in (Song & Eromn 2019), empirical analysis on the CIFAR-10 dataset would be needed --- if provided, my rating would increase.
- Bias and tradeoff analysis (similar to Song & Ermon 2019) is missing.

## Question:

- The hyperparameter $\alpha$ is said to be set to 1.0 (Section 3), which does not seem feasible based on the Equation (5) (7). Was it meant to be 0.5? Can the authors clarify on this? Also, I am curious how sensitive DEMI is on the choice of the prior hyperparameter $\alpha$. This would be a good analysis to have for the completeness of the paper.

## Additional comments:

- Section organization: I suggest having an introduction as a separate section, with methods being the following section. Section 3 (experiments) and 4 (results) can be combined. For section 2 (Related work), a different name could be considered because the main content here additionally includes a theoretical connection to existing approaches, which is in fact an important contribution of the paper.
- The plots in figure 2 are not properly scaled, with too many lines overlapping one another. I suggest the authors improve the plot for better readability.
- Typo in section 4.2: overalll.
- Please place a whitespace after the column (DEMI:) in the title.

---

> ### Author Response · Authors · 2020-11-25
> **Updated the discussion of related work and expanded experimental evaluation. Thank you for the feedback!**
>
> *1. The method only discusses estimation of mutual information, not maximization of MI for representation learning.*
>
> >Following the reviewers’ feedback, we performed the experiment of maximizing MI using different MI estimators including ours, based on Deep InfoMax (Hjelm et al., 2018). We found that DEMI outperforms the other MI estimators on the downstream CIFAR10 and CIFAR100 classification task. We describe this experiment in the updated manuscript.
>
> *2. The biggest weakness of this paper would be: experiments. The training data used in the experiments is either low-dimensional or synthetic, so there remains a question about how well this method will scale to a high-dimensional, and challenging deep learning setting. As in (Song & Eromn 2019), empirical analysis on the CIFAR-10 dataset would be needed --- if provided, my rating would increase.*
>
> >We thank the reviewer for the suggestion. We have performed a more “realistic” experiment on CIFAR10 and CIFAR100. We use different MI estimators including ours to perform representation learning, based on Deep InfoMax (Hjelm et al., 2018). We have found that DEMI outperforms the other MI estimators on the downstream CIFAR10 and CIFAR100 classification task. We have described this experiment in the updated manuscript.
>
> *3. Bias and tradeoff analysis (similar to Song & Ermon 2019) is missing.*
>
> >Our estimator relies on two approximations: (i) sample average estimate of the expectation, which is well understood and produces unbiased estimates of the expectation, and (ii) posterior probability prediction by a discriminative classifier. The ability of the classifier to match true log odds values in the test set in turn depends on the number of training sample pairs and the capacity of the neural architecture used in training. Thus the quality of our estimates is subject to the classical issues of model capacity and overfitting in deep learning. We leave theoretical analysis (which is closely related to the classifier’s convergence to the true separating boundary for a general hypothesis class) for future work.
>
> *4. The hyperparameter $\alpha$ is said to be set to 1.0 (Section 3), which does not seem feasible based on the Equation (5) (7). Was it meant to be 0.5? Can the authors clarify on this? Also, I am curious how sensitive DEMI is on the choice of the prior hyperparameter $\alpha$. This would be a good analysis to have for the completeness of the paper.*
>
> >Thank you for catching this typo! We fixed it in the paper.
>
> >As suggested by the reviewer, we ran DEMI with different settings of $\alpha$ ({0.25, 0.5, 0.75}) and reported the results in the updated manuscript.
>
> *5. Section organization: I suggest having an introduction as a separate section, with methods being the following section. Section 3 (experiments) and 4 (results) can be combined. For section 2 (Related work), a different name could be considered because the main content here additionally includes a theoretical connection to existing approaches, which is in fact an important contribution of the paper.*
>
> *The plots in figure 2 are not properly scaled, with too many lines overlapping one another. I suggest the authors improve the plot for better readability.*
>
> *Typo in section 4.2: overalll.*
>
> *Please place a whitespace after the column (DEMI:) in the title.*
>
> >We thank the reviewer for the suggestions! We carefully proofread and re-organized the paper, fixed typos, and improved the figures throughout the paper.

---

### Official Review · AnonReviewer5 · 2020-11-07
**A new classifier based discriminative mutual information estimator. Experiment is too simple and does not show really advantage.**

**Rating:** 4
**Confidence:** 4

**Review:**

This work suggests a new discriminative mutual information estimator that relies on a classifier to directly estimate the log density ratio of p(x,y)/p(x)p(y), without variational lower bound. In general, the idea is easy to follow and simple simulations are done to demonstrate its effectiveness. However, I still have some concerns:
1. A classifier based MI estimator reminds me of a closely related problems: the independence test. For the latter, there are also a few recent proposals based on a classifier to distinguish p(x,y) from p(x)p(y). I understand the methodologies are different, but I still feel some motivations are similar. It would be better if authors can clarify this point.
[1] Lopez-Paz, David, and Maxime Oquab. "Revisiting classifier two-sample tests." ICLR 2017.
[2] Sen, Rajat, Ananda Theertha Suresh, Karthikeyan Shanmugam, Alexandros G. Dimakis, and Sanjay Shakkottai. "Model-powered conditional independence test." NeurIPS 2017.

2. Authors discussed the theoretical optimum of their estimator when the number of samples approching to infinity. In the simulations, it seems that the number of training samples is also very large (e.g., 160k). What will happen in case of moderate or small number of samples?

3. For me, the simulation on the self-consistency tests does not demonstrate a big advantage of DEMI, especially considering that a few competitors are not included (e.g., GM mentioned in [Song and Ermon, 2019]). On the other hand, lots of work on variational MI (including this one) claim the great potential on representation learning with either mutual information maximization or information bottleneck. However, validations are totally missing. In this sense, it would be much better if authors can provide a simple representation learning demo, just like [Hjelm et al., 2018]. What will happen if we replce MINE with DEMI.

4. It seems from Fig. 1, the advantage of the estimator becomes more obvious with the increase of dimension. Can authors provide some explanation or theoretical analysis?

5. It seems to me the work is prepared in a quick time. There are a few typos (e.g., the 6 line in the second paragraph of page 3, $\hat{p}(x,y;\hat{D})$ should be $\hat{p}(y;\hat{D})$). The clarity and location of figures can be improved.

---

> ### Author Response · Authors · 2020-11-25
> **Updated the discussion of related work and expanded experimental evaluation. Thank you for the feedback!**
>
> *1. A classifier based MI estimator reminds me of a closely related problems: the independence test. For the latter, there are also a few recent proposals based on a classifier to distinguish p(x,y) from p(x)p(y). I understand the methodologies are different, but I still feel some motivations are similar. It would be better if authors can clarify this point. [1] Lopez-Paz, David, and Maxime Oquab. "Revisiting classifier two-sample tests." ICLR 2017. [2] Sen, Rajat, Ananda Theertha Suresh, Karthikeyan Shanmugam, Alexandros G. Dimakis, and Sanjay Shakkottai. "Model-powered conditional independence test." NeurIPS 2017.}*
>
> >We thank the reviewer for the comment. We agree these two problems are closely related and have expanded the Introduction to comment on this. Indeed, the independence test can be thought of as a binary hypothesis test for MI > 0 vs. MI=0. In this paper, we consider the associated task of estimating the MI value.
>
> *2. Authors discussed the theoretical optimum of their estimator when the number of samples approching to infinity. In the simulations, it seems that the number of training samples is also very large (e.g., 160k). What will happen in case of moderate or small number of samples?*
>
> >We thank the reviewer for this suggestion! We revised the paper to include results for several training set sizes (160k, 80k, 32k) and added a discussion of this point to Section 3 of the paper. Its performance relative to the other MI estimators held up with the training data size decreased.
>
> *3. For me, the simulation on the self-consistency tests does not demonstrate a big advantage of DEMI, especially considering that a few competitors are not included (e.g., GM mentioned in [Song and Ermon, 2019]). On the other hand, lots of work on variational MI (including this one) claim the great potential on representation learning with either mutual information maximization or information bottleneck. However, validations are totally missing. In this sense, it would be much better if authors can provide a simple representation learning demo, just like [Hjelm et al., 2018]. What will happen if we replce MINE with DEMI.*
>
> >Based on the reviewer’s feedback, we moved the self-consistency results to Appendix A in the updated manuscript. We agree that the self-consistency tests do not demonstrate a big advantage of DEMI but rather show that most MI estimators do well on such tests.
>
> >We thank the reviewer for the suggestion on providing a representation learning demo. We performed this experiment based on Deep InfoMax (Hjelm et al., 2018) by substituting different MI estimators including DEMI and MINE. We have found that DEMI outperforms MINE and the other MI estimators on the downstream CIFAR10 and CIFAR100 classification task. We describe this experiment in the updated manuscript.
>
> *4. It seems from Fig. 1, the advantage of the estimator becomes more obvious with the increase of dimension. Can authors provide some explanation or theoretical analysis?*
>
> >To be more precise, the improvements offered by our estimator are more pronounced for high values of MI, which is more easily realized in high dimensional settings. As we discuss in the Introduction, the estimates that rely on variational lower bounds are bounded from above by O(log N), where N is the number of sample pairs  (Song & Ermon, 2019; McAllester&Stratos, 2020). For the same number of training sample pairs, the estimators become increasingly inaccurate for high MI setups. The accuracy of our estimator is governed by the ability of the classifier to match true log odds values in the test set, which in turn depends on (i) the number of training sample pairs and (ii) the capacity of the neural architecture used in training. Thus the quality of our estimates is subject to the classical issues of model capacity and overfitting in deep learning. We leave theoretical analysis (which is closely related to the classifier’s convergence to the true separating boundary for a general hypothesis class) for future work. We also note that for low MI values relative to the size of the training set, our (and other) estimators could overestimate MI, as can be seen in the case of 20-dimensional Gaussian variables in Fig. 1. We expanded the Introduction and Experiments sections to comment on this.
>
> *5. It seems to me the work is prepared in a quick time. There are a few typos. The clarity and location of figures can be improved.*
>
> > We thank the reviewer for the suggestion. We carefully proofread the paper, fixed typos, and improved the figures throughout the paper.

---

### Official Review · AnonReviewer4 · 2020-11-08
**Good way to estimate mutual information**

**Rating:** 7
**Confidence:** 2

**Review:**

Seems like the most direct way to estimate mutual information using a classifier. I like this work because it is much more straight-forward than the prior work such as MINE. It shows sufficient performance on the experiments shown.

---

> ### Author Response · Authors · 2020-11-25
> **We agree! Thank you for the positive and encouraging review.**
>
> We agree! Thank you for the positive and encouraging review.

---

### Author Response · Authors · 2020-11-25
**Updated the discussion of related work and expanded experimental evaluation. Thank the reviewers!**

We thank the reviewers for their thoughtful feedback. Based on the reviewers' comments, we revised the paper to update the discussion of related work and expanded experimental evaluation to further demonstrate the advantages of our approach on real image data and in the context of representation learning.

We thank *R2* for bringing to our attention CCMI (Mukherjee et al., 2020), a recently demonstrated method that also employs the classification setup in the context of (conditional) MI estimation. We expanded our discussion of related methods and empirical evaluation to include CCMI. CCMI improves upon MINE by directly constructing the classifier function to be optimized over the training set (rather than optimizing over arbitrary functions), but it still employs the variational lower bound as the estimate of MI. In particular, the estimate relies both on paired and unpaired data. The term that depends on the unpaired data is asymptotically zero, but in the limited data setup leads to increased errors relative to our method (DEMI), as our experiments demonstrate.

*R2* observed that all methods tend to underestimate MI when it is high. Related, *R5* wondered why the advantages of our estimator become more apparent with increasing dimensionality of the data. To be more precise, the improvements offered by our estimator are more pronounced for high values of MI, which is more easily realized in high dimensional settings. As we discuss in the Introduction, the estimates that rely on variational lower bounds are bounded from above by O(log N), where N is the number of sample pairs (Song & Ermon, 2019; McAllester & Stratos, 2020). For the same number of training sample pairs, the estimators become increasingly inaccurate for high MI setups.

The accuracy of our estimator is governed by the ability of the classifier to match true log odds values in the test set, which in turn depends on (i) the number of training sample pairs and (ii) the capacity of the neural architecture used in training. *R3* requested we include bias/variance tradeoff analysis. DEMI uses two approximations: (i) sample average estimate of the expectation, which is well understood and produces unbiased estimates of the expectation, and (ii) posterior probability prediction by a discriminative classifier. The bias of our estimator can come from the classifier overfitting. Thus the quality of our estimates is subject to the classical issues of model capacity and overfitting in deep learning. We leave theoretical analysis (which is closely related to the classifier’s convergence to the true separating boundary for a general hypothesis class) for future work. We also note that for low MI values relative to the size of the training set, our (and other) estimators could overestimate MI, as can be seen in the case of 20-dimensional Gaussian variables in Fig. 1.

*R5* pointed out that our method seems closely related to the classification-based independence tests (Lopez-Paz & Oquab, 2017; Sen et al., 2017). We agree and have expanded the Introduction to comment on this. Indeed, the independence test can be thought of as a binary hypothesis test for MI > 0 vs. MI=0. In this paper, we consider the associated task of estimating the MI value.

Based on the reviewers’ feedback, we now include a demonstration of representation learning using DEMI in the paper. We performed this experiment based on Deep InfoMax (Hjelm et al., 2018) by substituting different MI estimators. We found that DEMI outperforms MINE and the other MI estimators on the downstream CIFAR10 and CIFAR100 classification task. In response to *R5*’s input, we moved the report on the self-consistency tests to Appendix A. Moreover, we included experiments on varying data sets sizes as requested by *R5* and on different settings of the hyperparameter $\alpha$ to address *R3*’s comments.

In addition, we carefully proofread the paper and fixed the typos identified by the reviewers and us, reorganized the structure of the sections as suggested by *R3*, and improved the readability of figures as requested by *R5*.

---

### Decision · Program_Chairs · 2021-01-07
**Final Decision**

**Decision:**

Reject

**Comment:**

In the paper, the authors propose a new method for estimating the mutual information based on a neural network classification that is fairly straight forward. The proposed method compares relatively well with known methods for estimating mutual information with a very large number of samples. The main issue of this classifier (a neural network) is that it requires that a classifier that discriminates between x, y pairs coming from p(x,y) and x, y pair coming from p(x)p(y) (this is done via reshuffling). The reviewers point out that the procedure is interesting, but it does not perform significantly better than the other proposed methods.

Also, I want to add that the proposed method is trained by using a given NN trained with 20 epochs and a mini-batch of 64. This is a significant issue because if we train the NN to reduce the validation error the posterior probability estimates are typically overconfident a significant work is being done to calibrate them. Why 20? How do we select this number if we cannot use a validation set? With less training example does 20 also work? This is very relevant because in the areas in which p(x,y)/p(x)p(y) is low for very high MI values getting these estimates correctly is critical. The classifier does not need to perform accurately in classification, but an estimation of the posterior probability and NNs will tend to be overconfident here and provide a biased estimate for these values. It will also provide an overestimate probability in the area that both p(x,y) and p(x)p(y) are high.

Finally, the authors reference the paper by Nguyen, Wainwright, and Jordan, but they do not acknowledge how that paper actually estimates log(p(x,y)/p(x)p(y)) similarly. That paper is very general and theoretical, and this paper can only be understood as a particular implementation of their solution. I think the authors missed that point in their paper. Also, I think the authors should acknowledge the papers that have come before using nearest neighbor or histograms for entropy estimation.